# Heavy-Ball Momentum Method in Continuous Time and Discretization Error Analysis

**Bochen Lyu[a,b,*]**    **Xiaojing Zhang[b]**    **Fangyi Zheng[c]**    **He Wang[d]**    **Zheng Wang[e]**

**Zhanxing Zhu[a*]**

[a]University of Southampton    [b]DataCanvas    [c]Pony.ai    [d]University College London
[e]University of Leeds
[*]{bochen.lyu, z.zhu}@soton.ac.uk

## Abstract

This paper establishes a continuous time approximation, a piece-wise continuous differential equation, for the discrete Heavy-Ball (HB) momentum method with explicit discretization error. Investigating continuous differential equations has been a promising approach for studying the discrete optimization methods. Despite the crucial role of momentum in gradient-based optimization methods, the gap between the original discrete dynamics and the continuous time approximations due to the discretization error has not been comprehensively bridged yet. In this work, we study the HB momentum method in continuous time while putting more focus on the discretization error to provide additional theoretical tools to this area. In particular, we design a first-order piece-wise continuous differential equation, where we add a number of counter terms to account for the discretization error explicitly. As a result, we provide a continuous time model for the HB momentum method that allows the control of discretization error to *arbitrary order* of the step size. As an application, we leverage it to find a new implicit regularization of the directional smoothness and investigate the implicit bias of HB for diagonal linear networks, indicating how our results can be used in deep learning. Our theoretical findings are further supported by numerical experiments.

## 1 Introduction

Gradient descent (GD) and its variants momentum methods, such as Polyak's Heavy-Ball momentum (HB) (Polyak, 1964), Nesterov's method of accelerated gradients (NAG) (Nesterov, 1983), and Adam (Kingma and Ba, 2017), are at the core of the success of training deep neural networks. This leads to the importance of understanding the gradient-based optimization methods, whereas the direct analysis for their discrete learning dynamics is challenging. Hence an optional approach is generally applied in this area: leveraging tools from dynamical systems to study the learning dynamics in continuous time to shed light on the dynamical behaviors of the discrete updates, e.g., Barrett and Dherin (2022); Lyu and Li (2020); Ji and Telgarsky (2019); Li et al. (2022); Lyu and Zhu (2023); Miyagawa (2023); Rosca et al. (2023). However, one fundamental gap—the discrepancy between the discrete optimizations and their continuous time models named as the *discretization error*—becomes manifest when employing this approach.

While recent works have made significant efforts to fill the aforementioned gap for GD (Barrett and Dherin, 2022; Miyagawa, 2023; Rosca et al., 2023), the analysis for the discretization error of the important variants of GD—momentum methods—in continuous time is still far from comprehensive. Along this line of research, Kovachki and Stuart (2020) demonstrated that the continuous time approximation of HB and NAG, which solves $\min_\beta L(\beta)$, can be expressed as a rescaled gradient

flow (RGF):

$$\dot{\beta} = -\frac{\nabla L(\beta)}{1 - \mu}, \tag{1}$$

where $L$ is the objective function and $\mu$ is the momentum factor. Despite its convenience for studying discrete HB and NAG, this approximation has one drawback: the insufficient consideration for discretization error when approaching the continuous time limit. As a result, it is insufficient to differentiate between HB and GD in continuous time. For example, the solutions of Eq. (1) and that of the gradient flow (GF) $\dot{\beta} = -\nabla L(\beta)$, the simplest continuous time model of GD, are not very different: if $\beta^*(t)$ solves RGF and $\beta^\dagger(t)$ solves GF, then $\beta^\dagger(t) = \beta^*((1 - \mu)t)$, which suggests that GD and HB will converge to similar points. However, this is inconsistent with the actual behavior, e.g., in Fig. 2(a) GD and HB converge to different points, which reveals that the RGF cannot fully capture the difference between HB and GD and motivates us to study continuous approximations for HB with smaller discretization error.

Recently, backward error analysis (Hairer et al., 2006) has been successfully applied to construct continuous time approximations for GD (Barrett and Dherin, 2022; Miyagawa, 2023; Rosca et al., 2023) with lower discretization error than GF. Inspired by these works, Ghosh et al. (2023) provided a continuous differential equation for HB, which achieves smaller discretization error compared to RGF (Eq. (1)). While promising, it still does not fully capture the actual discrete learning dynamics since the discretization error is only to the second order of the step size.

To this end, we also study the HB momentum method in continuous time while putting more focus on closing the gap due to the discretization error. In particular, we propose HB Flow (HBF), a pice-wise continuous differential equation as a novel continuous time approximation for the HB momentum method. We add a number of counter terms to the improved version of Eq. (1) to cancel the discretization error, obtaining a continuous time differential equation that can be *arbitrarily close* to HB, i.e., the discretization error can be controlled to arbitrary order of the step size. Our HBF provides a more reliable foundation for studying momentum methods in continuous time when the direct study of discrete learning dynamics is cumbersome. As a case study, we examine the implicit bias of HB, the preference for certain kind of solution without explicit regularization, by employing the HBF for the popular diagonal linear networks where there are already abundant results for GD (Azulay et al., 2021; Even et al., 2023; Pesme et al., 2021; Pillaud-Vivien et al., 2020; Woodworth et al., 2020; Yun et al., 2021).

**Contributions.** We study HB—one of the most important gradient-based optimization algorithms with momentum—in continuous time. In particular,

1. We propose a piece-wise continuous differential equation (HBF) by adding a counter term (Theorem 2.1) that can be expanded in a series formulation to different orders of the step size to the RGF (Eq. (1)), such that the HBF can precisely capture the learning dynamics of the discrete HB.

2. We explicitly show the leading order of the discretization error for the HBF and indicate the way how one can obtain HBF with a discretization error to any order of the step size (Section 2.2), revealing the explicit precision of HBF for approximating the discrete HB. These results are firstly revealed in this work to the best of our knowledge, bridging the gap between the discrete HB and the its precise characterization in continuous time.

3. We leverage our HBF to examine learning dynamics of HB. For example, we reveal that, as HBF implicitly has a regularization term for the directional smoothness, the learning dynamics of HB exhibits smaller directional smoothness compared to GD (Fig.1). In addition, for the application of HBF in deep learning, we investigate the implicit bias of HB for the diagonal linear network (Theorem 3.1) through the lens of HBF as a case study, revealing its difference compared to that of GF which cannot be obtained by the RGF.

## 1.1 Related Works

**Continuous approximation** Prior works (Shi et al., 2022; Su et al., 2016; Wilson et al., 2016) have constructed second-order ODEs to study the convergence properties of HB/NAG in continuous time, where the momentum factor typically depends on the step size and the iteration count. In this paper, we study HB with a constant momentum factor that is independent of step size and iteration

count, a setting that is more consistent with the practical case, e.g., PyTorch Paszke et al. (2019), and we focus more on the order of the discretization error of the continuous time model. Towards this direction, besides Kovachki and Stuart (2020); Ghosh et al. (2023), Cattaneo et al. (2023) focused on the continuous limit of a more general adaptive gradient-based methods with momentum, Adam, achieving a discretization error to the second order of the step size. As a comparison, we provide a continuous differential equation that can be arbitrarily close to the discrete HB.

**Implicit bias of optimizers** The implicit bias of GD for various deep neural networks has been widely studied in, e.g., Ji and Telgarsky (2019); Soudry et al. (2018); Yun et al. (2021) for linear networks and Chizat and Bach (2020); Lyu and Li (2020) for homogeneous networks. For diagonal linear networks studied in this paper, Azulay et al. (2021); Pesme et al. (2021); Pillaud-Vivien et al. (2020); Woodworth et al. (2020) revealed the interesting transition from kernel to rich regime by altering the scale of initialization. Papazov et al. (2024) studied HB using a second-order ordinary differential equation (ODE) with discretization error to the second order of step size. For linearly separable data and linear model, Wang et al. (2022) showed that momentum method converges to the $\ell_2$-max-margin solution, which is the same as GD. As a comparison, Zhang et al. (2024) revealed that Adam with negligible stability constant exhibits the preference of $\ell_\infty$-max-margin solution. Furthermore, Wang et al. (2021) studied the implicit bias of adaptive optimization algorithms on homogeneous deep neural networks.

## 1.2 Preliminaries

**Notations** For a vector $\beta \in \mathbb{R}^d$ that depends on time $t$, we use $\dot{\beta}$ and $\ddot{\beta}$ to denote its first and second derivative with respect to time $t$, respectively. We use $\beta_j$ to denote its $j$-th component and $\|\beta\|_p$ for its $\ell_p$-norm. We use $\alpha \cdot \beta$ to denote the inner product and $\odot$ to denote elementwise product, e.g., $\alpha \cdot A \cdot \beta$ denotes $\alpha^T A \beta$ for $\alpha \in \mathbb{R}^{d_1}, A \in \mathbb{R}^{d_1 \times d_2}, \beta \in \mathbb{R}^{d_2}$. We use $[N]$ for integers between $[0, N]$.

**Heavy-Ball momentum method** HB (Polyak, 1964) employs a two-step updating scheme (Sutskever et al., 2013), rather than the single-step manner of GD. Particularly, HB first accumulates the history of past iterations before updating the model parameter $\beta \in \mathbb{R}^d$, i.e., $m_{k+1} = \mu m_k - \eta \nabla L(\beta_k)$, $\beta_{k+1} = \beta_k + m_{k+1}$ where $\mu \in (0, 1)$ is the momentum factor, $\eta$ is the step size, $k$ is the iteration number, and $m \in \mathbb{R}^d$ is the momentum, which can be further written in a single equation

$$\text{Discrete HB:} \quad \beta_{k+1} - \beta_k = -\eta \nabla L(\beta_k) + \mu (\beta_k - \beta_{k-1}). \tag{2}$$

## 2 HB Momentum Methods in Continuous Time and Discretization Error

In this section, we will propose an ODE

$$\dot{\beta}(t) = -\mathcal{G}(\beta) \tag{3}$$

that can be arbitrarily close to the discrete learning dynamics of HB Eq. (33). To characterize the gap between the discrete learning dynamics and the ODE Eq. (3), given $N > 0$, we define the *discretization error* of Eq. (3) as

$$\forall k \in [N]: \ \varepsilon_k = \beta(k\eta) - \beta_k, \tag{4}$$

where $\beta(k\eta)$ solves Eq. (3) and we will denote $t_k = k\eta$ for convenience in the rest of this paper. An investigation of Euler forward method will identify $\eta$ as the step size. When $\varepsilon_k = \mathcal{O}(\eta^\alpha)$, we say that the discretization error of the continuous time model Eq. (3) is to the $\alpha$-th order of the step size. We aim to find the formulation of $\mathcal{G}(\beta)$ such that Eq. (3) can be arbitrarily close to the discrete HB, i.e., $\varepsilon_k = \mathcal{O}(\eta^\alpha)$ for any given $\alpha$.

**Intuition of our approach** Given the discrete HB Eq. (33), Kovachki and Stuart (2020) showed that $\varepsilon_k = \mathcal{O}(\eta)$ if $\mathcal{G}(\beta) = \nabla L/(1-\mu)$ as in Eq. (1). To coincide with such observation, our overall goal is to find the formulation of a modified ODE $\dot{\beta} = -G - \eta\gamma$ such that the discretization error can be controlled arbitrarily low, where its design should follow two basic principles: (i). it should coincide with RGF, the simplest continuous approximation of HB RGF, when $\eta$ is very small, thus, $G$ should degenerate to $\nabla L/(1-\mu)$; (ii). by adding the *counter term* $\gamma$, it should allow us to further

decrease the discretization error of RGF to get better continuous approximations for HB. To achieve this, we need to derive the exact forms of $G$ and $\gamma$ under the condition $\varepsilon_k = \mathcal{O}(\eta^\alpha)$. And there will be two key steps:

1. find the equations that $G$ and $\gamma$ must satisfy if we require the discretization error $\varepsilon_k = \mathcal{O}(\eta^\alpha)$ for any $\alpha > 0$;
2. solve these equations to give the formulations of $G$ and $\gamma$.

**Overview of our approach**  We now apply the above intuition to establish such a continuous time approximation of HB. Adding the counter term $\gamma$ directly to the RGF is, however, problematic due to the fact that each iteration of momentum methods exploits the history of previous iterations. This renders the local error analysis unreliable since it ignores previous updates. To perform a global analysis, instead of directly utilizing the backward error analysis in Barrett and Dherin (2022), we propose a piece-wise continuous differential equation by decomposing $\mathcal{G}(\beta)$ into two parts, named as HB Flow (HBF),

$$t \in [t_k, t_{k+1}): \ \dot{\beta} = -\mathcal{G}(\beta) := -G_k(\beta) - \eta\gamma_k(\beta). \tag{5}$$

This was previously discussed in Ghosh et al. (2023). In Eq. (5), $k$ denotes the iteration count for the discrete updates, $t_k = k\eta$, and $G_k$ depends on $k$ and should degenerate to $\nabla L/(1 - \mu)$ as in Eq. (1) for small step size. Additionally, the counter term $\gamma_k$ is designed to cancel the further discretization error brought by $G_k(\beta)$ such that $\varepsilon_k$ can be controlled to be arbitrarily small, hence the name counter term. As discussed in the above intuition, deriving the formulation of $\gamma_k$ needs a set of equations for it to satisfy. We establish such equations as follows. When approximating $t_k$ from $t > t_k$ and $t < t_k$, respectively, Taylor expansion for $\beta$ in Eq. (5) provides us

$$\begin{aligned} \beta(t_{k+1}) - \beta(t_k) &= \eta\dot{\beta}(t_k^+) + \eta^2 I_k^+ = -\eta G_k - \eta^2\gamma_k + \eta^2 I_k^+ \\ \beta(t_k) - \beta(t_{k-1}) &= \eta\dot{\beta}(t_k^-) - \eta^2 I_k^- = -\eta G_{k-1} - \eta^2\gamma_{k-1} - \eta^2 I_k^- \end{aligned} \tag{6}$$

where $t_k^+$ and $t_k^-$ mean that we approximate $t_k$ from $t > t_k$ and $t < t_k$, respectively, $\eta^2 I_k^\pm = \int_{t_k}^{t_{k\pm 1}} \ddot{\beta}(\tau)(t_{k\pm 1} - \tau)d\tau$, and we apply Eq. (5) for $\dot{\beta}(t_k^\pm)$. A subtraction of the discrete HB update Eq. (2) to the first equality of Eq. (6) now allows us to construct the relation between $\varepsilon_k$ and $\gamma_k$:

$$\varepsilon_{k+1} - \varepsilon_k = \mu(\varepsilon_k - \varepsilon_{k-1}) - \eta\left[G_k - \mu G_{k-1} - \nabla L(\beta_k)\right] + \eta^2\left[I_k^+ + \mu I_k^- - \gamma_k + \mu\gamma_{k-1}\right]. \tag{7}$$

As $I_k^\pm$ can be expressed by $\gamma_k$, requiring $\varepsilon_{k+1} - \varepsilon_k = \mathcal{O}(\eta^{\alpha+1})$ in Eq. (7) [1] immediately builds an equation for $\gamma_k$, which can be solved to give the form of $\gamma_k$. Hence, the discretization error of the continuous time model Eq. (5) for the discrete update Eq. (2) can be controlled to the $\alpha$-th order of the step size $\eta$. The formulation of Eq. (5) is presented in Theorem 2.1, and the detailed technical proofs are deferred to Appendix A.

## 2.1 HBF with Discretization Error to Arbitrary Order of the Step Size

Following the intuition and overview of our approach, we now solve Eq. (7) to derive $\gamma_k$ and $G_k$, and reveal that the obtained HBF by doing so is indeed an $\mathcal{O}(\eta^\alpha)$ (piece-wise) continuous time model of HB. As discussed earlier in Eq. (7), we need to find $\gamma_k$ to ensure that $\varepsilon_{k+1} - \varepsilon_k = \mathcal{O}\left(\eta^{\alpha+1}\right)$, hence the following integral functional equation

$$\eta^2(I_k^+ + \mu I_k^- - \gamma_k + \mu\gamma_{k-1}) = \mathcal{O}\left(\eta^{\alpha+1}\right) \tag{8}$$

must be satisfied. We solve this equation following three steps as shown below: **(1).** write the solution of Eq. (8), $\gamma_k$, as a series form

$$\gamma_k = \sum_{\sigma=0}^\infty \eta^\sigma \gamma_k^{(\sigma)}; \tag{9}$$

**(2).** derive $I_k^\pm$ explicitly by employing Eq. (9) in the learning dynamics Eq. (5) in the series form

$$I_k^+ = \sum_{\sigma=0}^\infty \eta^\sigma (\mathscr{I}_k^+)^{(\sigma)}; \tag{10}$$

---

[1] We will show in Appendix that this condition is sufficient for proving $\varepsilon_k = \mathcal{O}\left(\eta^\alpha\right)$ by induction.

**(3).** match the terms of Eq. (9) with that of Eq. (10) to each order of the step size $\eta$, i.e., $\forall \sigma \in \mathbb{N}$: $\gamma_k^{(\sigma)} = (\mathscr{I}_k^+)^{(\sigma)}$. As a result, a simple truncation of $\gamma_k$ will automatically lead Eq. (7) to give us $\varepsilon_{k+1} - \varepsilon_k = \mathcal{O}\left(\eta^{\alpha+1}\right)$ as desired. Below we formalize the aforementioned discussion.

**Theorem 2.1** (HBF with the discretization error $\varepsilon_k = \mathcal{O}\left(\eta^\alpha\right)$). *Let $k \in [N]$ be the iteration count, $\eta$ be the step size, and $t_k = k\eta$, the piece-wise continuous time differential equation HB Flow (HBF) for the discrete update Eq. (33)*

$$\text{HBF:} \quad \dot{\beta} = -G_k(\beta) - \eta\gamma_k(\beta), \quad t \in [t_k, t_{k+1}) \tag{11}$$

*has a discretization error that satisfies*

$$\varepsilon_{k+1} - \varepsilon_k = \mathcal{O}\left(\eta^{\alpha+1}\right) \tag{12}$$

*and, as a result, $\varepsilon_k = \mathcal{O}\left(\eta^\alpha\right)$ for $\alpha \geq 1$, if $G_k$ and $\gamma_k$ has the following formulations:*

$$G_k = \mu G_{k-1} + \nabla L \tag{13}$$

$$\gamma_k = \sum_{\sigma=0}^{\alpha-2} \eta^\sigma \gamma_k^{(\sigma)}. \tag{14}$$

*In particular, for ease of notation, let $\mathbf{L}_\beta^{(k,\sigma)} = \gamma_k^{(\sigma-1)} \cdot \nabla$ be a differential operator and*

$$\gamma_k^{(-1)} = G_k, \quad \mathcal{S}_{m,\sigma} = \{(\sigma_1, \ldots, \sigma_m) | \sum_{i=1}^{m} \sigma_i = \sigma - m + 2, \sigma_i \in \mathbb{N}\}$$

$$\chi_j^{(\sigma)} = \sum_{m=2}^{\sigma+2} \sum_{\mathcal{S}_{m,\sigma}} \frac{1}{m!}\left[(-1)^m \mathbf{L}_\beta^{(j,\sigma_1)} \cdots \mathbf{L}_\beta^{(j,\sigma_{m-1})} \gamma_j^{(\sigma_m-1)} + \mu \mathbf{L}_\beta^{(j-1,\sigma_1)} \cdots \mathbf{L}_\beta^{(j-1,\sigma_{m-1})} \gamma_{j-1}^{(\sigma_m-1)}\right],$$

*then each term of $\gamma_k$ in Eq. (14) can be simply written as*

$$\forall \sigma \in \mathbb{N}: \quad \gamma_k^{(\sigma)} = \sum_{j=0}^{k} \mu^{k-j} \chi_j^{(\sigma)}. \tag{15}$$

**Remark 1** Eq. (12) is an *equality* obtained from solving the integral functional equation Eq. (8), rather than an inequality bound. In addition, Eq. (12) is established to *arbitrary order of the step size* $\eta$ for any given $\alpha \geq 1$. Our approach is different from the previous continuous time model for HB method (Ghosh et al., 2023) where the discretization error is specifically constructed to the second order of the step size.

**Remark 2** In Eq. (13), the formulation of $G_k$ intuitively resembles the update of momentum in the discrete learning dynamics of HB $m_k = \mu m_{k-1} - \nabla L$. Interestingly, $G_k$ can be further simplified as

$$G_k = \frac{1 - \mu^{k+1}}{1 - \mu}\nabla L \xrightarrow{\text{large } k} \frac{\nabla L}{1 - \mu}, \tag{16}$$

which is exactly the R.H.S of the RGF (Eq. (1)). Moreover, Eq. (15) indicates that each iteration of HB depends on the history of previous iterations as $\gamma_k^{(\sigma)}$ incorporates information of all previous $\chi_j^{(\sigma)}$ with $j \leq k$, and such dependence decays very fast due to the coefficient $\mu^{k-j} \ll 1$ for small $j$. By letting $\mu = 0$, all the dependence on $k$ will disappear and our results can recover those of GD. Interestingly, it is worth mentioning that the difference between HBF and the continuous approximations of GD is closely related to the powers of $\eta(1 + \mu)/(1 - \mu)^2$ as we will show in Section 2.2.

**Remark 3** The formulations of $\gamma_k^{(\sigma)}$ are obtained in a recursive manner, ($\chi^{(\sigma)}$ is obtained from $\gamma_k^{(\sigma')}$'s with $\sigma' < \sigma$), hence Theorem 2.1 allows us to always build continuous time models for HB method with smaller discretization errors given those with larger discretization errors, instead of performing the whole set of analysis. For example, to build the HBF with $\varepsilon_k = \mathcal{O}\left(\eta^3\right)$ given HBF with $\varepsilon_k = \mathcal{O}\left(\eta^2\right)$, we only need to derive $\gamma_k^{(1)}$ (since $\gamma_k$ is truncated to the order $\alpha - 2 = 1$), which can be obtained by applying $\gamma_k^{(0)}$ provided by the HBF with $\varepsilon_k = \mathcal{O}\left(\eta^2\right)$. In addition, the recursive manner in Theorem 2.1 provides a possibility to calculate the involved terms automatically by using software for symbolic mathematics such as SymPy, which could be an interesting future work.

| $\varepsilon_k = \mathcal{O}(\eta^\alpha)$ | GD | HB |
|---|---|---|
| $\alpha = 1$ | $\dot{\beta} = -\nabla L$ | $\dot{\beta} = -\frac{\nabla L}{(1-\mu)}$ (Kovachki and Stuart, 2020) |
| $\alpha = 2$ | $\dot{\beta} = -\nabla L - \eta \frac{\nabla L \cdot \nabla^2 L}{2}$ (Barrett and Dherin, 2022) | $\dot{\beta} = -\frac{\nabla L}{1-\mu} - \eta \frac{1+\mu}{(1-\mu)^3} \frac{\nabla L \cdot \nabla^2 L}{2}$ Eq. (19) and (Ghosh et al., 2023) |
| $\alpha = 3$ | $\dot{\beta} = -\nabla L - \eta \frac{\nabla L \cdot \nabla^2 L}{2}$ $-\eta^2 \left[ \frac{\omega_1}{4} + \frac{\omega_2}{12} \right]$ Miyagawa (2023); Rosca et al. (2023) | $\dot{\beta} = -\frac{\nabla L}{1-\mu} - \eta \frac{1+\mu}{(1-\mu)^3} \frac{\nabla L \cdot \nabla^2 L}{2}$ $-\frac{\eta^2(1+\mu)^2}{(1-\mu)^5} \left[ \frac{\omega_1}{4} + \frac{(1+10\mu+\mu^2)\omega_2}{12(1+\mu)^2} \right]$ Eq. (22) of this work |
| Arbitrary $\alpha$ | Miyagawa (2023); Rosca et al. (2023) | Theorem 2.1 of this work |
| Discrete | $\beta_{k+1} = \beta_k - \nabla L(\beta_k)$ | $\beta_{k+1} = \beta_k - \eta \nabla L(\beta_k) + \mu(\beta_k - \beta_{k-1})$ |

Table 1: Continuous approximations for GD and HB up to different orders of discretization error.

## 2.2 HBF with Discretization Error $\varepsilon_k = \mathcal{O}(\eta^2)$ and $\varepsilon_k = \mathcal{O}(\eta^3)$

In this section, we derive HBF with discretization error to the second and third order of the step size to indicate how our approach works. There are basically three steps for finding a HBF with $\varepsilon_k = \mathcal{O}(\eta^\alpha)$ for $\alpha \geq 1$:

1. truncate $\gamma_k$ to the order $\alpha - 2$ in Eq. (14), i.e, $\gamma_k = \sum_{\sigma=0}^{\alpha-2} \gamma_k^{(\sigma)}$;

2. from the smallest $\sigma = 0$ to $\sigma = \alpha - 2$, find all $\chi_j^{(\sigma)}$ with $j \leq k$ by identifying the corresponding $\mathcal{S}_{m,\sigma}$ with $m = \{2, \ldots, \sigma + 2\}$ for each $\sigma$;

3. derive the expression of $\gamma_k^{(\sigma)}$ for all $\sigma \leq \alpha - 2$ in a recursive manner using the relation $\gamma_k^{(\sigma)} = \sum_{j=0}^k \mu^{k-j} \chi_j^{(\sigma)}$.

Below we discuss the cases for $\alpha = 2, 3$. We also summarize these results in Table 1. Note that the case for $\alpha = 1$ states that HBF is a RGF, i.e., $\dot{\beta} = -\nabla L/(1-\mu)$, which might not fully characterize the difference between momentum methods and vanilla GD.

### 2.2.1 HBF with $\varepsilon_k = \mathcal{O}(\eta^2)$

According to Theorem 2.1, there is only one term in the series of $\gamma_k$, i.e., $\gamma_k^{(0)}$. Recall that $\mathbf{L}_\beta^{(k,0)} = G_k \cdot \nabla$ and there is only one element in the set $\mathcal{S}_{m=2,\sigma=0}$, i.e., $\mathcal{S}_{m=2,\sigma=0} = \{(\sigma_1 = 0, \sigma_2 = 0)\}$, we obtain for $j \geq 1$:

$$\chi_j^{(0)} = \frac{1}{2} \left[ \mathbf{L}_\beta^{(j,0)} \gamma_j^{(-1)} + \mu \mathbf{L}_\beta^{(j-1,0)} \gamma_{j-1}^{(-1)} \right]. \tag{17}$$

Thus, using the definition of $\mathbf{L}_\beta^{(j,0)}$ and $\gamma_j^{(1)}$ according to Eq. (15) in Theorem 2.1, we can immediately derive that

$$\gamma_k = \gamma_k^{(0)} = \frac{\nabla L \cdot \nabla^2 L}{2(1-\mu)^2} \sum_{j=0}^k \mu^{k-j} \left[ (1 - \mu^{j+1})^2 + \mu(1 - \mu^j)^2 \right]. \tag{18}$$

Interestingly, all the dependence on the iteration count $k$ exists in the form of $\mu^k$, then for large iteration count $k$, the form of $\gamma_k$ can be largely simplified as $\gamma_k \approx \frac{1+\mu}{(1-\mu)^3} \frac{\nabla L \cdot \nabla^2 L}{2}$. This gives us HBF with $\varepsilon_k = \mathcal{O}(\eta^2)$:

$$\dot{\beta} = -\frac{\nabla L}{1-\mu} - \eta \frac{1+\mu}{(1-\mu)^3} \frac{\nabla L \cdot \nabla^2 L}{2}, \tag{19}$$

which is consistent with the $\mathcal{O}(\eta^2)$ continuous approximation of HB in Ghosh et al. (2023) while our derivation of HBF is in a different approach that does not depend inequality bounds. It is worth to mention that when $\mu = 0$, HBF recovers the $\mathcal{O}(\eta^2)$ continuous approximation of GD as expected.

### 2.2.2 HBF with $\varepsilon_k = \mathcal{O}(\eta^3)$

As shown in Theorem 2.1, in this case, the series of $\gamma_k$ should be truncated to the order $\alpha - 2 = 1$, hence $\gamma_k = \gamma_k^{(0)} + \eta \gamma_k^{(1)}$. Since we have already derived $\chi_j^{(0)}$ for HBF with $\varepsilon_k = \mathcal{O}(\eta^2)$ in

Section 2.2.1, we only need to find $\chi_k^{(1)}$, which will give us $\gamma_k^{(1)}$. We first find the collection of sets $\mathcal{S}_{m,\sigma}$, where $m = \{2, 3\}$ given $\sigma = 1$ as $m$ can be taken as integers between $[2, \sigma + 2]$ according to Theorem 2.1. Specifically, we have

$$
\begin{aligned}
\mathcal{S}_{2,1} &= \{(\sigma_1 = 1, \sigma_2 = 0), (\sigma_1 = 0, \sigma_2 = 1)\}, \\
\mathcal{S}_{3,1} &= \{(\sigma_1 = 0, \sigma_2 = 0, \sigma_3 = 0)\}.
\end{aligned}
\tag{20}
$$

We defer the rest of the detailed calculation to Appendix A and directly present the results for large $k$ below:

$$
\gamma_k^{(1)} = \frac{(1+\mu)^2}{4(1-\mu)^5}\left[\omega_1 + \frac{1 + 10\mu + \mu^2}{3(1+\mu)^2}\omega_2\right].
\tag{21}
$$

where we let

$$
\omega_1 = \left(\nabla L \cdot \nabla^2 L\right) \cdot \nabla^2 L, \quad \omega_2 = \nabla L \cdot \nabla\left(\nabla L \cdot \nabla^2 L\right).
$$

As a result, the HBF with $\varepsilon_k = \mathcal{O}\left(\eta^3\right)$ is

$$
\dot{\beta} = -\frac{\nabla L}{1-\mu} - \frac{\eta}{2}\frac{1+\mu}{(1-\mu)^3}\nabla L \cdot \nabla^2 L - \frac{\eta^2}{4}\frac{(1+\mu)^2}{(1-\mu)^5}\left[\omega_1 + \frac{1 + 10\mu + \mu^2}{3(1+\mu)^2}\omega_2\right].
\tag{22}
$$

**Implicit regularization of HBF** According to Eq. (19), HBF with $\varepsilon_k = \mathcal{O}\left(\eta^2\right)$ in Section 2.2.1 indicates that momentum induces a stronger implicit gradient regularization (IGR, Barrett and Dherin (2022)), i.e., $\gamma_{\mathrm{HB}} = (1+\mu)/(1-\mu)^3\gamma_{\mathrm{GD}}$ where $\gamma_{\mathrm{HB}}$ is the implicit regularization of HB while $\gamma_{\mathrm{GD}}$ is that of GD. For HBF with $\varepsilon_k = \mathcal{O}\left(\eta^3\right)$, we can conclude that the difference between HB and GD is more complicated since HBF now relies more on $\omega_2$ that primarily depends on $\nabla^3 L$. Interestingly, the formulation of HBF with $\alpha = 3$ suggests that HB will implicitly impose a regularization effect of *directional smoothness*, which is not the case for GD. In particular, for $\mu \approx 1$, the third term of Eq. (22) is close to

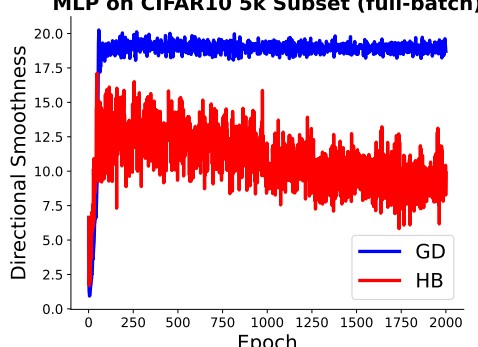

MLP on CIFAR10 5k Subset (full-batch)

$$
\omega_1 + \omega_2 = \nabla\left(\nabla L \cdot \left(\nabla^2 L \nabla L\right)\right),
\tag{23}
$$

which is an approximation for the directional smoothness (Ahn et al., 2022)

$$
\mathscr{D} = \frac{\nabla L(\beta) \cdot \left(\nabla L(\beta) - \nabla L(\beta - \eta\nabla L(\beta))\right)}{\eta\|\nabla L(\beta)\|^2}
$$

by expanding $\nabla L(\beta - \eta\nabla L(\beta))$ around $\beta$. The directional smoothness measures the extent of

Figure 1: Comparison of directional smoothness for HB and GD for MLP on CIFAR-10 with full-batch GD and HB ($\mu = 0.9$) with $\eta = 0.1$.

oscillating behavior of optimization algorithms, i.e., the discrepancy between two adjacent iterates. For GD, Ahn et al. (2022) revealed that it exhibits oscillatory behavior such that its directional smoothness, $\mathscr{D}^{\mathrm{GD}}$, would saturate around $2/\eta$. As a comparison, due to the implicit regularization for the directional smoothness of our HBF, HB prefers learning dynamics with smaller directional smoothness $\mathscr{D}^{\mathrm{HB}} < \mathscr{D}^{\mathrm{GD}}$, implying an oscillatory behavior to less extent compared to GD. This is verified in Fig. 1, where $\mathscr{D}^{\mathrm{GD}} \approx 2/\eta = 20$ while $\mathscr{D}^{\mathrm{HB}} < \mathscr{D}^{\mathrm{GD}}$ and keeps decreasing. The numerical experimental details can be found in Appendix C.

## 3 Implicit Bias of Momentum Methods through HBF

The HBF proposed in Theorem 2.1 provides a reliable mathematical tool for analyzing a wide variety of properties of HB. One crucial aspect is its implicit bias in deep learning. To demonstrate the significance of HBF, we characterize the implicit bias of HBF specifically for the two layer diagonal linear network (Woodworth et al., 2020) as a case study.

**The formulation of 2-layer diagonal linear networks** A 2-layer diagonal linear network (DLN) with parameter $\mathbf{w} = (\mathbf{w}_+, \mathbf{w}_-)$ where $\mathbf{w}_\pm \in \mathbb{R}^d$ is equivalent to a linear predictor

$$f(x; \mathbf{w}) = x^T \mathbf{w} := x^T (\mathbf{w}_+ \odot \mathbf{w}_+ - \mathbf{w}_- \odot \mathbf{w}_-), \qquad (24)$$

where we use the parameterization $\mathbf{w} = \mathbf{w}_+ \odot \mathbf{w}_+ - \mathbf{w}_- \odot \mathbf{w}_-$ (Woodworth et al., 2020). This model is a popular proxy model for deep neural networks as it shares many interesting phenomena with more complex architectures, e.g., the transition from kernel to rich regime. In this section, we focus on this model with $\mathbf{w}_{+;j}(0) = \mathbf{w}_{-;j}(0)$.

**Leaning task** For our task, given a dataset $\{(x_i, y_i)\}_{i=1}^n$ with $n$ samples where $x_i \in \mathbb{R}^d$ and $y_i \in \mathbb{R}$, we assume that $n < d$ and consider the regression problem with quadratic loss $L(\mathbf{w}_+, \mathbf{w}_-) = \sum_{i=1}^n (x_i^T \mathbf{w} - y_i)^2 / (2n)$. We use $X \in \mathbb{R}^{n \times d}$ to denote the data matrix and $y = (y_1, \ldots, y_n)^T \in \mathbb{R}^n$.

**Implicit bias of GF for diagonal linear networks** For GF, Azulay et al. (2021); Woodworth et al. (2020) showed that the limit point of $\mathbf{w}$ is equivalent to the solution of the constrained optimization problem $\mathbf{w}(\infty) = \arg\min_{\mathbf{w}} \Lambda^{\mathrm{GF}}(\mathbf{w}; \kappa(0))$, $s.t.\ X\mathbf{w} = y$ where $\kappa_j(0) = \mathbf{w}_{+;j}(0)\mathbf{w}_{-;j}(0)$, the potential function $\Lambda^{\mathrm{GF}}(\mathbf{w}; \kappa(0)) = \sum_{j=1}^d \Lambda_j^{\mathrm{GF}}(\mathbf{w}; \kappa(0))$, and

$$\Lambda_j^{\mathrm{GF}}(\mathbf{w}; \kappa(0)) = \frac{1}{4} \left[ \mathbf{w}_j \operatorname{arcsinh}\left( \frac{\mathbf{w}_j}{2\kappa_j(0)} \right) - \sqrt{4(\kappa_j(0))^2 + \mathbf{w}_j^2} + 2\kappa_j(0) \right]. \qquad (25)$$

Note that $\kappa(0)$ controls the transition from rich regime to kernel regime, i.e., $\Lambda^{\mathrm{GF}}(\mathbf{w}; \kappa(0)) \to \|\mathbf{w}\|_1$ for small $\kappa(0)$ while $\Lambda^{\mathrm{GF}}(\mathbf{w}; \kappa(0)) \to \|\mathbf{w}\|_2$ large $\kappa(0)$ (Woodworth et al., 2020).

### 3.1 Implicit Bias of HBF for Diagonal Linear Networks

According to Theorem 2.1, the learning dynamics of the diagonal linear networks $f(x; \mathbf{w})$ can be written as

$$\dot{\mathbf{w}}_+ = -\frac{\nabla_{\mathbf{w}_+} L}{1 - \mu} - \eta \gamma^{\mathbf{w}_+}, \quad \dot{\mathbf{w}}_- = -\frac{\nabla_{\mathbf{w}_-} L}{1 - \mu} - \eta \gamma^{\mathbf{w}_-} \qquad (26)$$

where we use $\gamma^{\mathbf{w}_+} \in \mathbb{R}^d$ and $\gamma^{\mathbf{w}_-} \in \mathbb{R}^d$ for HBF of $\mathbf{w}_+$ and HBF of $\mathbf{w}_-$, respectively, and we use $\gamma_{;j}^{\mathbf{w}_\pm}$ to denote its $j$-th component. Compared to RGF (Eq. (1)), Eq. (26) has one extra term that accounts for the high-order discretization error. The implicit bias of $\mathbf{w}$ under the RGF is similar to that of GF, which, however, is not the case for Eq. (26).

**Theorem 3.1** (Implicit bias of HBF for diagonal linear networks)**.** *If the dynamics of diagonal linear network $f(x; \mathbf{w}) = x^T \mathbf{w}$ where $\mathbf{w} = \mathbf{w}_+ \odot \mathbf{w}_+ - \mathbf{w}_- \odot \mathbf{w}_-$ follows HBF defined in Theorem 2.1 and if $\mathbf{w}(\infty)$ converges to an interpolation solution, let $\kappa_j(t) = \mathbf{w}_{+;j}(0)\mathbf{w}_{-;j}(0)\exp(-\eta\epsilon_j(t))$ where $\epsilon_j(t) = \int_0^t ds\, (\gamma_{;j}^{\mathbf{w}_+}(s)/\mathbf{w}_{+;j}(s) + \gamma_{;j}^{\mathbf{w}_-}(s)/\mathbf{w}_{-;j}(s))$, then $\mathbf{w}(\infty)$ satisfies that*

$$\mathbf{w}(\infty) = \arg\min_{\mathbf{w}} \Lambda(\mathbf{w}; \kappa) \quad s.t.\ X\mathbf{w} = y, \qquad (27)$$

*where $\Lambda(\mathbf{w}; \kappa) = \sum_{j=1}^d \Lambda_j(\mathbf{w}, t = \infty; \kappa(\infty))$ with*

$$\Lambda_j(\mathbf{w}, t; \kappa(t)) = \Lambda_j^{\mathrm{GF}}(\mathbf{w}; \kappa(t)) + \mathbf{w}_j \varphi_j(t), \quad \varphi_j(t) = \frac{\eta}{4} \int_0^t ds \left( \frac{\gamma_{;j}^{\mathbf{w}_+}(s)}{\mathbf{w}_{+;j}(s)} - \frac{\gamma_{;j}^{\mathbf{w}_-}(s)}{\mathbf{w}_{-;j}(s)} \right). \qquad (28)$$

**Comparison with the implicit bias of GF** Compared to the implicit bias of GF in Eq. (25), there are two differences brought by the high-order correction terms of HBF: **(1)**. the potential function $\Lambda_j^{\mathrm{GF}}(\mathbf{w}; \kappa(0))$ for GF becomes $\Lambda_j^{\mathrm{GF}}(\mathbf{w}; \kappa(\infty))$ for HBF where $\kappa(\infty)$ is different from $\kappa(0)$, meaning that HBF induces an effect equivalent to a rescaling of the initialization; **(2)**. $\Lambda_j(\mathbf{w}, \infty; \kappa)$ additionally depends on the product $\mathbf{w}_j \varphi_j(\infty)$. A similar term will appear in the potential function of GF $\Lambda^{\mathrm{GF}}$ if the initialization no longer satisfies $\mathbf{w}_+(0) = \mathbf{w}_-(0)$. In this sense, HBF also brings an effect that is equivalent to breaking the symmetry of the initialization. Theorem 3.1 also applies to the case for higher-order continuous approximation of GD by setting $\mu = 0$, suggesting an effect equivalent to the rescaling of the initialization that has been verified in GD (Even et al., 2023). This further reveals the reliability and usefulness of high-order continuous approximations.

The comprehensive characterization for the HBF requires a detailed investigation for the formulations of $\gamma^\pm$ specifically for the diagonal linear networks, which will be an open problem, while below we focus on $\varepsilon_k = \mathcal{O}\left(\eta^2\right)$ as an example.

**Corollary 3.2** (Implicit bias of HBF for diagonal linear networks with $\varepsilon_k = \mathcal{O}\left(\eta^2\right)$). *Under conditions of Theorem 3.1, if we use HBF with $\varepsilon_k = \mathcal{O}\left(\eta^2\right)$, then*

$$\kappa_j(t) = \kappa_j(0) \exp\left[\frac{\eta(1+\mu)}{(1-\mu)^2}\left(-\frac{\Phi_j(t)}{1-\mu} + \frac{\left(X^T X \mathbf{q}\right)_j}{n}\right)\right] \tag{29}$$

*where $\Phi_j(t) = 4\int_0^t ds(\partial_{\mathbf{w}_j} L)^2 > 0$, $\mathbf{q} \in \mathbb{R}^d$ with $\mathbf{q}_i = \sqrt{\mathbf{w}_i^2(\infty) + 4\kappa_i^2(0)} - 2\kappa_i(0)$.*

For the exponent of $\kappa_j(\infty)$, when $\Phi_j(\infty) > 0$ dominates, e.g., $\kappa_i(0) \gg \mathbf{w}_i(\infty)$, we will conclude that $\kappa_j(\infty) < \kappa_j(0)$, hence the rescaling effect brought by HBF equivalently reduces the initialization. Therefore, compared to $\Lambda^{\mathrm{GF}}$, $\Lambda(\mathbf{w}; \kappa)$ will closer to the $\ell_1$-norm and the solution $\mathbf{w}(\infty)$ will enjoy better sparsity. This finding is consistent with parts of results in Papazov et al. (2024), which analyzed the implicit bias of HB also with a continuous time differential equation that is a second-order ODE, while our HBF is a first-order ODE and can also cover the case for GD simply by letting $\mu = 0$.

## 4    Numerical Experiments

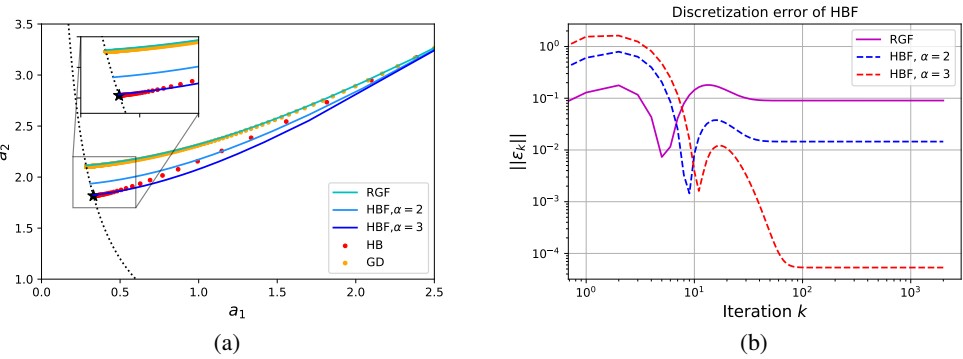

Figure 2: (a). Trajectories for learning dynamics of GD, HB, RGF, and HBF with discretization error $\mathcal{O}\left(\eta^2\right)$ and $\mathcal{O}\left(\eta^3\right)$ in a 2-d model. All dynamics start from the same point $(a_1 = 2.8, a_2 = 3.5)$. The convergence point of HB is denoted as a black star. The black dotted line denotes the set of all global minima. (b). Discretization errors for different continuous approximations of HB during training in (a).

In this section, we show numerical experiments on a simple 2-d model to verify our theoretical claims, and we present numerical experiments details and more experiments for diagonal linear networks in Appendix C.

Our simple 2-d model has the formulation $f(x; a_1, a_2) = a_1 a_2 x$, where $a_1, a_2 \in \mathbb{R}$ are the model parameters and $x, y \in \mathbb{R}$ is the training data. The loss function is $L = (f(x; a_1, a_2) - y)^2/2$. All parameters $a_1, a_2$ satisfying $a_1 a_2 x = y$ are global minima. To show that higher-order HBFs with discretization error $\mathcal{O}(\eta^\alpha)$ are better approximations for HB, we visualize trajectories for different learning dynamics, i.e., GD, HB, RGF, HBF with $\alpha = 2$, and HBF with $\alpha = 3$, in Fig. 2(a). The trajectory of HBF with $\alpha = 3$ is closer to that of HB than both RGF and HBF with $\alpha = 2$. Furthermore, Fig. 2(a) also reveals that RGF is more similar to GD and it cannot capture the discrete learning dynamics of HB well. We also plot the norm of discretization errors $\|\varepsilon_k\|^2$ for these continuous approximations during training in Fig. 2(b), where HBF with $\alpha = 3$ has the lowest discretization error after several steps. These results validate the reliability of HBF as a proxy of HB.

## 5 Conclusion

In this paper, we have established a new continuous time model for the discrete HB method (Eq. (2)), namely HBF, with an explicit discretization error that can be controlled to arbitrary order of the step size $\eta$. In particular, our approach constructs a relation, which is a functional integral equation, between discretization errors of adjacent iterates for any step and can be solved to arbitrary order of the step size. This is a different approach compared to prior works (Kovachki and Stuart, 2020; Ghosh et al., 2023). Our results provide a reliable foundation for analyzing the momentum methods in the continuous time limit. We leverage our HBF to shed lights on a newly observed implicit regularization effect of the HB method: the preference for small directional smoothness compared to GD. In addition, as another interesting application of our HBF in deep learning, we study the implicit bias of HBF for the popular proxy model diagonal linear networks, and we reveal the difference between the implicit bias of HB and that of GD which cannot be captured by RGF.

**Limitation and future directions**   The framework in this paper does not consider optimization methods with adaptive learning rate, e.g., Adam (Kingma and Ba, 2017). A generalization of our framework to such case would be an interesting future direction. In addition, our analysis can be generalized to the stochastic case by replacing $\nabla L$ with $\tilde{\nabla} L$, the approximate stochastic gradient, by following Li et al. (2018); Latz (2021). Finally, we only consider the simple diagonal linear networks in this paper, and future works can explore more complex deep learning models with our HBF to study the implicit bias of HB.

## Acknowledgments and Disclosure of Funding

B.L. is funded by a studentship provided by the School of Electronics and Computer Science, University of Southampton. The authors acknowledge DataCanvas AlayaNeW for providing computational resources. The authors thank the insightful and constructive feedback from the anonymous reviewers.

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

# Appendix

- In Appendix A, we provide proofs for Section 2.

- In Appendix B, we present proofs of Section 3.

- In Appendix C, we show details of numerical experiments in Section 2.2.2 and 4, and present more related numerical experiments to support our theoretical claims.

## A    Proofs for Section 2

We prove Theorem 2.1 in Appendix A.2 and give the details for the first several orders HBF in A.3. We first discuss the conditions that guarantee HBF as an effective approximation of HB.

### A.1    Conditions of the Effectiveness of HBF

To make the HBF a valid continuous approximation of HB, there are two necessary conditions:

1. It is crucial to control the ratio between $\eta$ and $1 - \mu$ to avoid $\eta \gg 1 - \mu$, which might lead the Taylor series to diverge. More interestingly, we conjecture that it is the magnitude of a special composite quantity

$$\psi := \frac{\eta}{(1-\mu)^2} \tag{30}$$

   that matters for the effectiveness of the HBF. This quantity spontaneously appears in both HBF with $\alpha = 2$ and $\alpha = 3$ but not in the RGF, i.e., for HBF with $\alpha = 2$ the counter term is proportional to $\psi$ while for HBF with $\alpha = 3$ a new counter term proportional to $\psi^2$ will appear. If $\psi$ is too large, then our results would no longer hold.

   Hence, we need to fix the value of $\mu$ and treat only $\eta$ as the variable to denote the higher-order terms as $\mathcal{O}(\eta^\alpha)$ while hide $\mu$ in the expansion. And it would be interesting for future works to study the case when both $\mu$ and $\eta$ are treated as variables such that higher-order terms are denoted as $\mathcal{O}(\psi^\alpha)$ for $\alpha \geq 1$. In addition, in the regime of large $\mu$ and large $\eta$, the model might not be trained properly either: the update direction coming from the gradient and that from the momentum will jointly affect the training direction significantly, while these two directions can be very different due to the large value of $\mu$ and $\eta$ hence cannot give a consistent updating direction.

   In addition, the dependence of HBF on the special composite quantity $\psi$ is consistent with the empirical observation in Leclerc and Madry (2020), where the optimization curves for different momentum values can be recovered by a corresponding change in the learning rate. The dependence of HB on $\eta$ and $\mu$ at the same time further indicates the advantage of HBF with $\alpha > 1$ and that RGF, which only depends on $\mu$, is not sufficient to reflect the optimization properties of HB.

2. Given $\alpha$, the continuous approximations include derivatives of $L$ up to the $\alpha$-th order, hence $L$ should at least be $\alpha$-times continuously differentiable and $||\nabla^\alpha L||$ should be upper bounded.

### A.2    Proof of Theorem 2.1

Given the HBF for $k \in [N]$

$$t \in [t_k, t_{k+1}] : \quad \dot{\beta} = -G_k(\beta) - \eta \gamma_k(\beta) \tag{31}$$

with unknown $G_k$ and $\gamma_k$, we expect that the counter term $\gamma_k$ could cancel higher-order discretization errors and $G_k$ should degenerate to rescaled gradient, i.e., $\nabla L/(1-\mu)$. Hence, $\gamma_k$ and $G_k$ should be designed in such a way that $\beta(t_k)$ is close to $\beta_k$ in the sense that the discretization error

$$\varepsilon_k = \beta(t_k) - \beta_k \tag{32}$$

is small. Inspired by Miyagawa (2023), we first present the outline of our three main steps for deriving their formulations below:

$$\boxed{\text{Step I}} \qquad\qquad \text{Unknown } G_k,\ \gamma_k \xrightarrow{\text{determine}} \text{Taylor expansion residual } I_k^{\pm}$$

$$\searrow \qquad \swarrow$$

$$\boxed{\text{Step II}} \quad \left\{ \begin{array}{c} \text{Taylor Expansion of } \beta(t_{k\pm1})\ (35),\ (36) \\ \downarrow \text{constructs} \\ \text{Expression of } \varepsilon_{k+1} - \varepsilon_k\ (41) \\ \downarrow \text{ required to be } \mathcal{O}(\eta^{\alpha+1}) \text{ for } \varepsilon_k = \mathcal{O}(\eta^{\alpha})\ (\text{Lemma A.2}) \\ \text{Equalities for } G_k,\ \gamma_k\ (44) \end{array} \right.$$

$$\downarrow \text{ solved by matching to each order of } \eta$$

$$\boxed{\text{Step III}} \qquad\qquad \text{Solution of } G_k \text{ and } \gamma_k$$

We now discuss the detailed proof following this outline.

*Proof.* We start with Step I, which deals with how $I_k^{\pm}$ is determined by $G_k$ and $\gamma_k$.

**Step I**   Recall that the discrete learning dynamics of HB is

$$\beta_{k+1} - \beta_k = -\eta\nabla L(\beta_k) + \mu(\beta_k - \beta_{k-1}), \tag{33}$$

where $\mu$ is the momentum factor and $k$ is the iteration number. Based on our discussion in Section 2, the continuous differential equation for HB is

$$\dot{\beta} = -G_k(\beta) - \eta\gamma_k(\beta) \tag{34}$$

for $t \in [t_k, t_{k+1})$ where $t_k = k\eta$ and the solution is $\beta(t)$. For arbitrary unknown $\gamma_k$ in Eq. (34), $I_k^{\pm}$ is determined, which is also unknown but depends on $\gamma_k$. Specifically, the first-order Taylor expansion with the remainder term in the integral form gives us

$$\beta(t_{k+1}) - \beta(t_k) = \eta\dot{\beta}(t_k^+) + \eta^2 I_k^+ = -\eta G_k - \eta^2\gamma_k + \eta^2 I_k^+ \tag{35}$$

where $t_k^+$ means we approximate $t_k$ from $t > t_k$,

$$I_k^+ = \int_0^1 \ddot{\beta}\left(\eta(k+s)\right)(1-s)ds,$$

and we use Eq. (34) in the second equality. Similarly, when approximating $t_k$ from $t < t_k$, we obtain that

$$\beta(t_k) - \beta(t_{k-1}) = -\eta G_{k-1} - \eta^2\gamma_{k-1} - \eta^2 I_k^-. \tag{36}$$

Now we construct the dependence of $I_k^{\pm}$ on $\gamma_k$ in the series form, as shown in Lemma. A.1 (proof can be found in Appendix A.2.1.).

**Lemma A.1.** *Given the series form*

$$\gamma_k = \sum_{\sigma=0}^{\infty} \eta^{\sigma}\gamma_k^{(\sigma)} \tag{37}$$

*and the continuous time differential equation Eq. (34), $I_k^{\pm}$ in Eq. (35) and Eq. (36) have the following series forms:*

$$I_k^+ = \sum_{p=0}^{\infty} \eta^p(\mathscr{I}_k^+)^{(p)} := \sum_{p=0}^{\infty}\sum_{q=2}^{p+2}\sum_{\sum_{j=1}^q \sigma_j = p-q+2} \frac{(-1)^q}{q!}\eta^p \mathbf{L}_{\beta}^{(k,\sigma_1)}\cdots\mathbf{L}_{\beta}^{(k,\sigma_{q-1})}\gamma_k^{(\sigma_q-1)}, \tag{38}$$

$$I_k^- = \sum_{p=0}^{\infty} \eta^p(\mathscr{I}_k^-)^{(p)} := \sum_{p=0}^{\infty}\sum_{q=2}^{p+2}\sum_{\sum_{j=1}^q \sigma_j = p-q+2} \frac{1}{q!}\eta^p \mathbf{L}_{\beta}^{(k-1,\sigma_1)}\cdots\mathbf{L}_{\beta}^{(k-1,\sigma_{q-1})}\gamma_{k-1}^{(\sigma_q-1)}. \tag{39}$$

**Step II**   Given the dependence of $I_k^\pm$ on $\gamma_k$, we are now able to write the Taylor expansion of $\beta(t_{k\pm 1})$ explicitly. This allows us to construct a relation between $\varepsilon_{k+1}$ and $\varepsilon_k$ by subtracting Eq. (33) from both sides of Eq. (35):

$$\varepsilon_{k+1} - \varepsilon_k = -\eta\left[G_k(\beta(t_k)) - \nabla L(\beta_k)\right] - \eta^2\gamma_k + \eta^2 I_k^+ - \mu(\beta_k - \beta_{k-1}). \tag{40}$$

Note that

$$\beta_k - \beta_{k-1} = \beta(t_k) - \beta(t_{k-1}) - (\varepsilon_k - \varepsilon_{k-1})$$
$$= -\eta G_{k-1} - \eta^2\gamma_{k-1} - \eta^2 I_k^- - (\varepsilon_k - \varepsilon_{k-1}),$$

we obtain the expression of $\varepsilon_{k+1} - \varepsilon_k$ as in the Step II of our outline:

$$\varepsilon_{k+1} - \varepsilon_k = \mu(\varepsilon_k - \varepsilon_{k-1}) - \eta\left[G_k - \mu G_{k-1} - \nabla L(\beta(t_k) - \varepsilon_k)\right]$$
$$+ \eta^2\left[I_k^+ + \mu I_k^- - \gamma_k + \mu\gamma_{k-1}\right]. \tag{41}$$

Eq. (41) builds the connection between the discretization error $\varepsilon_k$ and the counter term $\gamma_k$ in the continuous time differential equation Eq. (5). We can now construct the equalities for $G_k$ and $\gamma_k$ under the constraint of low discretization error by following the lemma below (proof can be found in Appendix A.2.2).

**Lemma A.2.** *For the continuous differential equation Eq.* (34) *and the discrete sequence given by Eq.* (33), *if*

$$G_k(\beta(t_k)) = \mu G_{k-1}(\beta(t_k)) + \nabla L(\beta(t_k)) \tag{42}$$

*with $G_{-1} = 0$ and*

$$I_k^+ + \mu I_k^- - \gamma_k + \mu\gamma_{k-1} = \mathcal{O}\left(\eta^{\alpha - 1}\right), \tag{43}$$

*as in Eq.* (49), *then we have*

$$\varepsilon_k - \varepsilon_{k-1} = \mathcal{O}\left(\eta^{\alpha + 1}\right)$$

*and, as a result,*

$$\varepsilon_k = \mathcal{O}\left(\eta^\alpha\right).$$

As shown in Lemma. A.2, to ensure that the leading order of the L.H.S of Eq. (41) is to the order of $\alpha > 1$, we only need to require $\varepsilon_{k+1} - \varepsilon_k = \mathcal{O}\left(\eta^\alpha\right)$ which can be guaranteed by Eq. (42) and the functional integral equation Eq. (43). As Eq. (42) can be easily solved by induction, we only need to solve Eq. (43). To achieve this, we build a stronger functional equation below:

$$I_k^+ + \mu I_k^- = \gamma_k - \mu\gamma_{k-1}, \tag{44}$$

which is the final equation that we aim to solve to derive $\gamma_k$.

**Step III**   The Step III of our outline is solving the functional integral equation Eq. (44). Our core idea is simple: as the series form of $I_k^\pm$ has already derived in Lemma. A.1, we make Eq. (44) satisfied by matching both sides of it for each order of $\eta$. In particular, given the series forms of $I_k^\pm$ in Lemma A.1 and that of $\gamma_k$ (Eq. (37)), we require

$$\forall p \in \mathbb{N}: \ \eta^p(\gamma_k^{(p)} - \mu\gamma_{k-1}^{(p)}) = \eta^p\left((\mathscr{I}_k^+)^{(p)} + \mu(\mathscr{I}_k^-)^{(p)}\right), \tag{45}$$

which, let

$$\chi_k^{(\sigma)} = (\mathscr{I}_k^+)^{(\sigma)} + \mu(\mathscr{I}_k^-)^{(\sigma)}, \tag{46}$$

gives us the recursive relation of $\gamma_k^{(\sigma)}$ for $\sigma \in \mathbb{N}$

$$\gamma_k^{(\sigma)} = \mu\gamma_{k-1}^{(\sigma)} + \chi_k^{(\sigma)} \tag{47}$$

because $\chi_k^{(\sigma)}$ only depends on $\gamma_k^{(\sigma')}$ with $\sigma' < \sigma$ according to Lemma A.1. Now given $\alpha \in \mathbb{Z}^+$, we can truncate $\gamma_k$ to the order $\alpha - 2$ such that

$$\gamma_k = \sum_{\sigma=0}^{\alpha-2} \eta^\sigma\gamma_k^{(\sigma)}, \tag{48}$$

then the functional integral equation is solved to the $(\alpha - 2)$-th order of the step size $\eta$:

$$
\begin{aligned}
I_k^+ + \mu I_k^- - \gamma_k + \mu\gamma_{k-1} &= \sum_{\sigma=0}^{\infty} \eta^\sigma \chi_k^{(\sigma)} - \sum_{\sigma=0}^{\alpha-2} \eta^\sigma \left( \gamma_k^{(p)} - \mu\gamma_{k-1}^{(p)} \right) \\
&= \sum_{\sigma=\alpha-1}^{\infty} \eta^\sigma \chi_k^{(\sigma)} = \mathcal{O}\left( \eta^{\alpha-1} \right),
\end{aligned}
\tag{49}
$$

which is exactly the condition in Lemma. A.2. Therefore, by constructing $\gamma_k$ following Eq. (47) and truncating $\gamma_k$ to preserver its first $\alpha - 2$ terms, we can prove that the discretization error of the continuous time differential equation Eq. (34) is to the order $\mathcal{O}\left(\eta^\alpha\right)$. $\qquad\square$

### A.2.1  Proof for Lemma A.1

*Proof.* We first rewrite $I_k^\pm$ as follows :

$$
\begin{aligned}
I_k^\pm &= \frac{1}{\eta^2} \int_{k\eta}^{k\eta\pm\eta} \ddot{\beta}(\tau)(k\eta \pm \eta - \tau)d\tau \\
&\stackrel{\tau' \leftarrow \tau - k\eta}{=} \frac{1}{\eta^2} \int_0^{\pm\eta} \left[ \sum_{n=0}^{\infty} \frac{1}{n!} \frac{d^n}{dt^n} \ddot{\beta}(k\eta)\tau'^n \right]^\pm (\pm\eta - \tau')d\tau' \\
&= \sum_{n=0}^{\infty} \frac{(\pm\eta)^n}{(n+2)!} \frac{d^n}{dt^n} \ddot{\beta}(t_k^\pm)
\end{aligned}
\tag{50}
$$

where we use $\int_0^\eta \tau'^n(\eta - \tau')d\tau' = \frac{\eta^{n+2}}{n+1} - \frac{\eta^{n+2}}{n+2} = \frac{\eta^{n+2}}{(n+1)(n+2)}$ in the last equality. To continue, we need the expression of $d^n\beta/dt^n$ and we start with $t \to t_k^+$:

$$
\begin{aligned}
\frac{d^n}{dt^n}\beta(t_k^+) &= \frac{d}{dt}\left( \frac{d^{n-1}}{dt^n}\beta(t_k^+) \right) \\
&= \dot{\beta}(t_k^+) \cdot \nabla \left( \frac{d^{n-1}}{dt^n}\beta(t_k^+) \right) \\
&= -(G_k + \eta\gamma_k) \cdot \nabla \left( \frac{d^{n-1}}{dt^n}\beta(t_k^+) \right) \\
&= (-1)^n (\mathbf{L}_\beta^{(k)})^{n-1} (G_k + \eta\gamma_k)
\end{aligned}
\tag{51}
$$

where we denote the differential operator $\mathbf{L}_\beta^{(k)} = (G_k + \eta\gamma_k) \cdot \nabla$ and use Eq. (34) in the third equality. Now suppose that $\gamma_k$ can be written as a series

$$
\gamma_k = \sum_{\sigma=0}^{\infty} \eta^\sigma \gamma_k^{(\sigma)}, \quad \gamma_k^{(-1)} = G_k,
$$

then Eq. (51) becomes

$$
\begin{aligned}
\frac{d^n}{dt^n}\beta(t) &= (-1)^n \left( \sum_{\sigma_1=0}^{\infty} \eta^{\sigma_1}\gamma_k^{(\sigma_1-1)} \cdot \nabla \right) \cdots \\
&\quad \cdots \left( \sum_{\sigma_{n-1}=0}^{\infty} \eta^{\sigma_{n-1}}\gamma_k^{(\sigma_{n-1}-1)} \cdot \nabla \right) \left( \sum_{\sigma_{n-1}=0}^{\infty} \eta^{\sigma_n}\gamma_k^{(\sigma_n-1)} \right) \\
&= (-1)^n \sum_{\sigma_1,\ldots,\sigma_n=0}^{\infty} \eta^{\sum_{j=1}^n \sigma_j} \mathbf{L}_\beta^{(k,\sigma_1)} \cdots \mathbf{L}_\beta^{(k,\sigma_{n-1})} \gamma_k^{(\sigma_n-1)}
\end{aligned}
\tag{52}
$$

Combined with Eq. (50), we obtain the form of $I_k^+$ for $t \in (t_k, t_{k+1})$ as

$$
\begin{aligned}
I_k^+ &= \sum_{n=0}^{\infty} \frac{\eta^n}{(n+2)!} \frac{d^{n+2}}{dt^{n+2}} \beta(t) \\
&= \sum_{n=0}^{\infty} \sum_{\sigma_1,\dots,\sigma_{n+2}=0}^{\infty} \frac{(-1)^{n+2}}{(n+2)!} \eta^{n+\sum_{j=1}^{n+2}\sigma_j} \mathbf{L}_\beta^{(k,\sigma_1)} \cdots \mathbf{L}_\beta^{(k,\sigma_{n+1})} \gamma_k^{(\sigma_{n+2}-1)} \\
&= \sum_{n=0}^{\infty} \sum_{m=0}^{\infty} \sum_{\sum_{j=1}^{n+2}\sigma_j=m} \frac{(-1)^{n+2}}{(n+2)!} \eta^{n+m} \mathbf{L}_\beta^{(k,\sigma_1)} \cdots \mathbf{L}_\beta^{(k,\sigma_{n+1})} \gamma_k^{(\sigma_{n+2}-1)} \\
&= \sum_{p=0}^{\infty} \sum_{q=2}^{p+2} \sum_{\sum_{j=1}^{q}\sigma_j=p-q+2} \frac{(-1)^q}{q!} \eta^p \mathbf{L}_\beta^{(k,\sigma_1)} \cdots \mathbf{L}_\beta^{(k,\sigma_{q-1})} \gamma_k^{(\sigma_q-1)}
\end{aligned}
\tag{53}
$$

where we let $p \leftarrow n + \sum_{j=1}^{n+2} \sigma_j, q \leftarrow n+2$, in the last equality. Similarly, when $t \to t_k^-$, we have

$$
\frac{d^n}{dt^n} \beta(t_k^-) = (-1)^n (\mathbf{L}_\beta^{(k-1)})^{n-1} (G_{k-1} + \eta \gamma_{k-1})
$$

which implies that

$$
I_k^- = \sum_{p=0}^{\infty} \sum_{q=2}^{p+2} \sum_{\sum_{j=1}^{q}\sigma_j=p-q+2} \frac{1}{q!} \eta^p \mathbf{L}_\beta^{(k-1,\sigma_1)} \cdots \mathbf{L}_\beta^{(k-1,\sigma_{q-1})} \gamma_{k-1}^{(\sigma_q-1)}.
\tag{54}
$$

$\square$

### A.2.2  Proof for Lemma A.2

*Proof.* We first present several useful relations. As Eq. (49) is established by solving the functional equation for any iteration count $k$, we can write the relation between $\varepsilon_{k+1}$ and $\varepsilon_k$ Eq. (41) as

$$
\varepsilon_{k+1} - \varepsilon_k = \mu(\varepsilon_k - \varepsilon_{k-1}) - \eta \left[ \nabla L\left( \beta(t_k) \right) - \nabla L\left( \beta(t_k) - \varepsilon_k \right) \right] + \mathcal{O}\left( \eta^{\alpha+1} \right).
\tag{55}
$$

If $\varepsilon_k = \mathcal{O}(\eta^\alpha)$, then Eq. (55) implies

$$
\begin{aligned}
\|\varepsilon_{k+1} - \varepsilon_k\| &\le \mu \|\varepsilon_k - \varepsilon_{k-1}\| + \eta \|\nabla L\left( \beta(t_k) \right) - \nabla L\left( \beta(t_k) - \varepsilon_k \right)\| + c_1 \eta^{\alpha+1} \\
&\le \mu \|\varepsilon_k - \varepsilon_{k-1}\| + \eta \lambda \|\varepsilon_k\| + c_1 \eta^{\alpha+1}
\end{aligned}
\tag{56}
$$

for some constant $c_1$ where we use $G_k - \mu G_{k-1} = \nabla L$ and let $\lambda = \max_\beta \|\nabla^2 L(\beta)\|$.

Denoting

$$
\forall k : c_2(k) = \frac{c_1}{\lambda} e^{\frac{2\lambda\eta}{1-\mu}k}, \; c_3(k) = \frac{2c_1}{1-\mu} e^{\frac{2\lambda\eta}{1-\mu}k},
$$

we now prove by induction.

1. For the first step ($k = 0$), by definition we have

$$
\varepsilon_0 = 0 \le c_2(0)\eta^\alpha.
$$

Note that $G_{-1} = 0$ and $\gamma_{-1} = 0$ by definition, then we have

$$
\|\varepsilon_1 - \varepsilon_0\| \le c_1 \eta^{\alpha+1} \le \frac{2c_1}{1-\mu} = c_3(0)\eta^{\alpha+1}
$$

since $\mu < 1$.

2. Suppose that for the $k$-th step the following relations hold:

$$
\begin{aligned}
\|\varepsilon_k\| &\le c_2(k)\eta^\alpha, \\
\|\varepsilon_{k+1} - \varepsilon_k\| &\le c_3(k)\eta^{\alpha+1}.
\end{aligned}
\tag{57}
$$

Then for the $(k+1)$-th step, we have

$$\begin{aligned}
\|\varepsilon_{k+1}\| &= \|\varepsilon_{k+1} - \varepsilon_k + \varepsilon_k\| \\
&\leq \|\varepsilon_{k+1} - \varepsilon_k\| + \|\varepsilon_k\| \\
&\leq c_2(k)\left(1 + \eta\frac{c_3(k)}{c_2(k)}\right)\eta^\alpha \\
&\leq c_2(k)e^{\frac{2\lambda\eta}{1-\mu}}\eta^\alpha \\
&= c_2(k+1)\eta^\alpha
\end{aligned}$$

where the last inequality is because $e^x > 1 + x$ for $x > 0$. Similarly,

$$\begin{aligned}
\|\varepsilon_{k+2} - \varepsilon_{k+1}\| &\leq \mu\|\varepsilon_{k+1} - \varepsilon_k\| + \eta\lambda\|\varepsilon_{k+1}\| + c_1\eta^{\alpha+1} \\
&\leq [\mu c_3(k) + \lambda c_2(k+1) + c_1]\eta^{\alpha+1} \\
&= \left[\mu e^{-\frac{2\lambda\eta}{1-\mu}} + \frac{1-\mu}{2} + \frac{1-\mu}{2}e^{-\frac{2\lambda\eta}{1-\mu}(k+1)}\right]c_3(k+1)\eta^{\alpha+1} \\
&\leq \left[\mu + \frac{1-\mu}{2} + \frac{1-\mu}{2}\right]c_3(k+1)\eta^{\alpha+1} \\
&= c_3(k+1)\eta^{\alpha+1}.
\end{aligned}$$

$\square$

## A.3   $\mathcal{O}(\eta^\alpha)$-close HBF for a specific $\alpha$

In this section, we derive the form of $\mathcal{O}(\alpha)$-close HBF for given a specific $\alpha$. There are basically three steps to find a HBF that is $\mathcal{O}(\eta^\alpha)$-close to HB:

1. truncate $\gamma_k$ to the desired order $\alpha$, i.e, $\gamma_k = \sum_{\sigma=0}^{\alpha-2}\gamma_k^{(\sigma)}$;

2. from the smallest $\sigma$, find all $\chi_j^{(\sigma)}$ with $j \leq k$ by finding the corresponding $\mathcal{S}_{m,\sigma}$ with $m = \{2, \ldots, \sigma+2\}$ for each $\sigma$;

3. derive the expression of $\gamma_k^{(\sigma)}$ for all $\sigma \leq \alpha - 2$ in a recursive manner using the relation $\gamma_k^{(\sigma)} = \sum_{j=0}^k \mu^{k-j}\chi_j^{(\sigma)}$.

In the following, we give the cases for $\alpha = 2$ and $3$ as examples. With this approach, one can in fact find HBF with arbitrary order of closeness to HB.

### A.3.1   $\alpha = 2$.

According to Theorem 2.1, the series of $\gamma_k$ is truncated to the first term, i.e., $\gamma_k = \eta^0\gamma_k^{(0)}$, where $\gamma_k = \sum_{j=0}^k \mu^{k-j}\chi_j^{(0)}$. Thus the first step is to find $\chi_j^{(0)}$, which can be given by first identifying the set $\mathcal{S}$:

$$\mathcal{S}_{m=2,\sigma=0} = \{(\sigma_1 = 0, \sigma_2 = 0)\}, \tag{58}$$

therefore there is only one term in $\chi_j^{(0)}$:

$$\chi_j^{(0)} = \frac{1}{2}\left[\mathbf{L}_\beta^{j,0}\gamma_j^{(-1)} + \mu\mathbf{L}_\beta^{j-1,0}\gamma_{j-1}^{(-1)}\right].$$

Recall that

$$\gamma_j^{(-1)} = G_j = \frac{1 - \mu^{j+1}}{1 - \mu}\nabla L, \tag{59}$$

which, according to our definition in Theorem 2.1, leads to

$$\mathbf{L}_\beta^{j,0} = \gamma_j^{(-1)} \cdot \nabla = G_j \cdot \nabla,$$

we obtain that

$$\chi_j^{(0)} = \frac{1}{2} \left[ G_j \cdot \nabla G_j + \mu G_{j-1} \cdot \nabla G_{j-1} \right]$$

$$= \frac{1}{2(1-\mu)^2} \left[ (1 - \mu^{j+1})^2 + \mu(1 - \mu^j)^2 \right] \nabla L \cdot \nabla^2 L. \tag{60}$$

Thus

$$\gamma_k^{(0)} = \frac{1}{2} \sum_{j=0}^{k} \mu^{k-j} \left[ G_j \cdot \nabla G_j + \mu G_{j-1} \cdot \nabla G_{j-1} \right]$$

$$= \frac{\nabla L \cdot \nabla^2 L}{2(1-\mu)^2} \sum_{j=0}^{k} \mu^{k-j} \left[ (1 - \mu^{j+1})^2 + \mu(1 - \mu^j)^2 \right]$$

$$= \frac{\nabla L \cdot \nabla^2 L}{2(1-\mu)^2} \sum_{j=0}^{k} \left[ (1 + \mu)\mu^{k-j} + \mu^{k+1}(\mu^j(1 + \mu) - 4) \right]. \tag{61}$$

When $k$ is larege, the above expression can be simplified as

$$\gamma_k^{(0)} \approx \frac{(1+\mu)\sum_{j=0}^{k} \mu^j}{2(1-\mu)^2} \nabla L \cdot \nabla^2 L \approx \frac{1+\mu}{2(1-\mu)^3} \nabla L \cdot \nabla^2 L.$$

### A.3.2  $\alpha = 3$.

Similarly, in this case we first truncate the series of $\gamma_k$ to the desired order, i.e., $\gamma_k = \gamma_k^{(0)} + \eta \gamma_k^{(1)}$ where we have already obtained $\gamma_k^{(0)}$ in the last section, thus we only need to find $\gamma_k^{(1)}$ and $\chi_k^{(1)}$, which can be done by first finding the set $\mathcal{S}_{m=2,\sigma=1}$ and $\mathcal{S}_{m=3,\sigma=1}$:

$$\mathcal{S}_{2,1} = \{ (\sigma_1 = 1, \sigma_2 = 0), (\sigma_1 = 0, \sigma_2 = 1) \},$$
$$\mathcal{S}_{3,1} = \{ (\sigma_1 = 0, \sigma_2 = 0, \sigma_3 = 0) \}.$$

Therefore there are three terms of $\chi_j^{(1)}$:

$$\chi_j^{(1)} = \frac{1}{2} \left[ \mathbf{L}_\beta^{j,1} \gamma_j^{(-1)} + \mu \mathbf{L}_\beta^{j-1,1} \gamma_{j-1}^{(-1)} \right] + \frac{1}{2} \left[ \mathbf{L}_\beta^{j,0} \gamma_j^{(0)} + \mu \mathbf{L}_\beta^{j-1,0} \gamma_{j-1}^{(0)} \right]$$

$$- \frac{1}{6} \left[ \mathbf{L}_\beta^{j,0} \mathbf{L}_\beta^{j,0} \gamma_j^{(-1)} - \mu \mathbf{L}_\beta^{j-1,0} \mathbf{L}_\beta^{j-1,0} \gamma_{j-1}^{(-1)} \right]. \tag{62}$$

Recall that $\gamma_j^{(-1)} = G_j$, $\mathbf{L}_\beta^{j,0} = G_j \cdot \nabla$, and $\mathbf{L}_\beta^{j,1} = \gamma_j^{(0)} \cdot \nabla$, the first line of Eq. (62) is

$$\frac{1}{2} \left[ \gamma_j^{(0)} \cdot \nabla G_j + \mu \gamma_{j-1}^{(0)} \cdot \nabla G_{j-1} + G_j \cdot \nabla \gamma_j^{(0)} + \mu G_{j-1} \cdot \nabla \gamma_{j-1}^{(0)} \right] \tag{63}$$

while the second line is

$$- \frac{1}{6} \left[ G_j \cdot \nabla (G_j \cdot \nabla G_j) - \mu G_{j-1} \cdot \nabla (G_{j-1} \cdot \nabla G_{j-1}) \right]. \tag{64}$$

To simplify these terms, we can either replace all $\gamma_j^{(0)}$ with the expression in Eq. (61) and write $G_j$ explicitly, or notice the recursive relation between $G_j$ and $G_{j-1}$ in Theorem 2.1, i.e. ,$G_j = \mu G_{j-1} + \nabla L$, then Eq. (63) becomes

$$\frac{1}{2} \left[ \gamma_j^{(0)} \cdot \nabla^2 L + \nabla L \cdot \nabla \gamma_j^{(0)} \right] + \frac{\mu}{2} \left[ \left( \gamma_j^{(0)} + \gamma_{j-1}^{(0)} \right) \cdot \nabla G_{j-1} + G_{j-1} \cdot \nabla \left( \gamma_j^{(0)} + \gamma_{j-1}^{(0)} \right) \right]$$

and Eq. (64) is now

$$- \frac{1}{6} \nabla L \cdot \nabla (G_j \cdot \nabla G_j) - \frac{\mu}{6} G_{j-1} \cdot \nabla (G_j \cdot \nabla G_j - G_{j-1} \cdot \nabla G_{j-1}). \tag{65}$$

Summing over these terms gives us $\chi_j^{(1)}$:

$$\chi_j^{(1)} = \Psi_j^{(1)} + \mu \Theta_j^{(1)} \tag{66}$$

where

$$\Psi_j^{(1)} = \frac{1}{2}\left(\gamma_j^{(0)} \cdot \nabla^2 L + \nabla L \cdot \nabla \gamma_j^{(0)}\right) - \frac{1}{6}\nabla L \cdot \nabla \left(G_j \cdot \nabla G_j\right)$$

$$\Theta_j^{(1)} = \frac{1}{2}\left[\left(\gamma_j^{(0)} + \gamma_{j-1}^{(0)}\right) \cdot \nabla G_{j-1} + G_{j-1} \cdot \nabla \left(\gamma_j^{(0)} + \gamma_{j-1}^{(0)}\right)\right]$$
$$- \frac{1}{6}G_{j-1} \cdot \nabla \left(G_j \cdot \nabla G_j - G_{j-1} \cdot \nabla G_{j-1}\right).$$

We can now find $\gamma_k^{(1)}$ through its definition:

$$\gamma_k^{(1)} = \sum_{j=0}^{k} \mu^{k-j}\chi_j^{(1)} = \sum_{j=0}^{k} \mu^{k-j}\Psi_j^{(1)} + \mu \sum_{j=0}^{k} \mu^{k-j}\Theta_j^{(1)}. \tag{67}$$

In the following, we derive the form of $\gamma_k^{(1)}$ When $k$ is large. According to Eq. (61), we have

$$\mu^{k-j}\gamma_j^{(0)} = \mu^{k-j}\frac{\nabla L \cdot \nabla^2 L}{2(1-\mu)^2}\sum_{i=0}^{j}\left[(1+\mu)\mu^{j-i} + \mu^{j+1}(\mu^i(1+\mu) - 4)\right]$$

$$= \mu^{k-j}\frac{\nabla L \cdot \nabla^2 L}{2(1-\mu)^2}\left[\frac{(1+\mu)(1-\mu^{j+1})}{1-\mu} + \frac{\mu^{j+1}(1+\mu)(1-\mu^{j+1})}{1-\mu} - 4(j+1)\mu^{j+1}\right]$$

$$= \frac{\nabla L \cdot \nabla^2 L}{2(1-\mu)^2}\left[\frac{(1+\mu)(\mu^{k-j} - \mu^{k+1})}{1-\mu} + \frac{\mu^{k+1}(1+\mu)(1-\mu^{j+1})}{1-\mu} - 4(j+1)\mu^{k+1}\right]$$

$$= \frac{\nabla L \cdot \nabla^2 L}{2(1-\mu)^2}\left[\frac{(1+\mu)\mu^{k-j}}{1-\mu} - \frac{\mu^{k+j+1}(1+\mu)}{1-\mu} - 4(j+1)\mu^{k+1}\right]$$

$$\approx \mu^{k-j}\frac{(1+\mu)}{2(1-\mu)^3}\nabla L \cdot \nabla^2 L \tag{68}$$

and, according to Eq. (59),

$$\mu^{k-j}G_j \cdot \nabla G_j = \frac{\mu^{k-j}(1 - 2\mu^{j+1} + \mu^{2(j+1)})}{(1-\mu)^2}\nabla L \cdot \nabla^2 L \approx \frac{\mu^{k-j}}{(1-\mu)^2}\nabla L \cdot \nabla^2 L. \tag{69}$$

Combining Eq. (68) and (69) gives the form of $\mu^{k-j}\Psi_j^{(1)}$ when $k$ is large:

$$\mu^{k-j}\Psi_j^{(1)} \approx \frac{\mu^{k-j}(1+\mu)}{4(1-\mu)^3}\left[(\nabla L \cdot \nabla^2 L) \cdot \nabla^2 L + \nabla L \cdot \nabla \left(\nabla L \cdot \nabla^2 L\right)\right]$$

$$- \frac{\mu^{k-j}}{6(1-\mu)^2}\nabla L \cdot \nabla \left(\nabla L \cdot \nabla^2 L\right)$$

which immediately leads to

$$\sum_{j=0}^{k} \mu^{k-j}\Psi_j^{(1)}$$

$$\approx \frac{(1+\mu)}{4(1-\mu)^4}\left[(\nabla L \cdot \nabla^2 L) \cdot \nabla^2 L + \nabla L \cdot \nabla \left(\nabla L \cdot \nabla^2 L\right)\right] - \frac{\nabla L \cdot \nabla \left(\nabla L \cdot \nabla^2 L\right)}{6(1-\mu)^3}$$

$$= \frac{1}{4(1-\mu)^4}\left[(1+\mu)(\nabla L \cdot \nabla^2 L) \cdot \nabla^2 L + \frac{(1+5\mu)}{3}\nabla L \cdot \nabla \left(\nabla L \cdot \nabla^2 L\right)\right]. \tag{70}$$

The left part is now deriving the form of $\mu^{k-j}\Theta_j^{(1)}$, which can be done by first finding

$$\mu^{k-j}\gamma_j^{(0)} \cdot \nabla G_{j-1} \approx \mu^{k-j}\frac{(1+\mu)}{2(1-\mu)^3}(\nabla L \cdot \nabla^2 L) \cdot \nabla G_{j-1}$$

$$\approx \mu^{k-j}\frac{(1+\mu)}{2(1-\mu)^4}(\nabla L \cdot \nabla^2 L) \cdot \nabla^2 L \approx \mu^{k-j}\gamma_{j-1}^{(0)} \cdot \nabla G_{j-1} \tag{71}$$

and

$$\mu^{k-j}G_{j-1}\cdot\nabla\left(G_j\cdot\nabla G_j - G_{j-1}\cdot\nabla G_{j-1}\right) \approx \frac{2\mu^{k-j}}{(1-\mu)^3}\nabla L\cdot\nabla\left(\nabla L\cdot\nabla^2 L\right), \qquad (72)$$

thus

$$\sum_{j=0}^{k}\mu^{k-j}\Theta_j^{(1)} \approx \frac{(1+\mu)}{2(1-\mu)^5}\left[(\nabla L\cdot\nabla^2 L)\cdot\nabla^2 L + \nabla L\cdot\nabla\left(\nabla L\cdot\nabla^2 L\right)\right]. \qquad (73)$$

Combing this equation with Eq. (70), we can now conclude the form of $\gamma_k^{(1)}$ when $k$ is large:

$$\gamma_k^{(1)} = \sum_{j=0}^{k}\mu^{k-j}\left(\Psi_j^{(1)} + \mu\Theta_j^{(1)}\right)$$

$$\frac{1}{4(1-\mu)^4}\left[(1+\mu)(\nabla L\cdot\nabla^2 L)\cdot\nabla^2 L + \frac{(1+5\mu)}{3}\nabla L\cdot\nabla\left(\nabla L\cdot\nabla^2 L\right)\right]$$

$$+ \frac{\mu(1+\mu)}{2(1-\mu)^5}\left[(\nabla L\cdot\nabla^2 L)\cdot\nabla^2 L + \nabla L\cdot\nabla\left(\nabla L\cdot\nabla^2 L\right)\right]$$

$$= \frac{(1+\mu)^2}{4(1-\mu)^5}\left[(\nabla L\cdot\nabla^2 L)\cdot\nabla^2 L + \frac{1+10\mu+\mu^2}{3(1+\mu)^2}\nabla L\cdot\nabla\left(\nabla L\cdot\nabla^2 L\right)\right] \qquad (74)$$

Note that when $\mu = 0$ we recover the result of GD, i.e., $\gamma_k^{(1)} = \frac{(\nabla L\cdot\nabla^2 L)\cdot\nabla^2 L}{4} + \frac{\nabla L\cdot\nabla\left(\nabla L\cdot\nabla^2 L\right)}{12}$.

## B  Proofs for Section 3

Given data $(x_i, y_i)$, the architecture of 2-layer diagonal linear network is

$$f(x_i; \mathbf{w}) = x_i^T(\mathbf{w}_+ \odot \mathbf{w}_+ - \mathbf{w}_- \odot \mathbf{w}_-) = \sum_{j=1}^{d} x_{i;j}\left(\mathbf{w}_{+;j}^2 - \mathbf{w}_{-;j}^2\right) \qquad (75)$$

and the empirical loss function is

$$L(\mathbf{w}) = \frac{1}{2n}\sum_{i=1}^{n}(f(x_i; \mathbf{w}) - y_i)^2.$$

We let $r = (r_1, \ldots, r_n)^T \in \mathbb{R}^n$ be the residual where $\forall i : r_i = f(x_i; \mathbf{w}) - y_i$. According to Theorem 2.1, the HBF learning dynamics of model parameters $\mathbf{w}_+$ and $\mathbf{w}_-$ will be

$$\dot{\mathbf{w}}_+ = -\frac{\nabla_{\mathbf{w}_+}L}{1-\mu} - \eta\gamma_k^{\mathbf{w}_+}, \quad \dot{\mathbf{w}}_- = -\frac{\nabla_{\mathbf{w}_-}L}{1-\mu} - \eta\gamma_k^{\mathbf{w}_-} \qquad (76)$$

where we use $\gamma_k^{\mathbf{w}_+} \in \mathbb{R}^d$ and $\gamma_k^{\mathbf{w}_-} \in \mathbb{R}^d$ to represent the error terms for HBF of $\mathbf{w}_+$ and $\mathbf{w}_-$, respectively, and the gradients are

$$\nabla_{\mathbf{w}}L = \frac{1}{n}X^T r, \qquad (77)$$

$$\nabla_{\mathbf{w}_+}L = 2\mathbf{w}_+ \odot \nabla_{\mathbf{w}}L, \quad \nabla_{\mathbf{w}_-}L = -2\mathbf{w}_- \odot \nabla_{\mathbf{w}}L. \qquad (78)$$

Using the expressions above, it can be easily verified that

$$\mathbf{w}_- \odot \nabla_{w_+}L + \mathbf{w}_+ \odot \nabla_{w_-}L = 0, \qquad (79)$$

and we will frequently use this relation later. Recall the definition of $\kappa_j = \mathbf{w}_{+;j}\mathbf{w}_{-;j}$, we now present useful lemmas before proving Theorem 3.1.

**Lemma B.1.** *Let* $\kappa_j(t) = \mathbf{w}_{+;j}(t)\mathbf{w}_{-;j}(t)$, $\gamma_{k;j}^{\mathbf{w}_\pm}$ *denote the $j$-th component of* $\gamma_k^{\mathbf{w}_\pm}$, *and*

$$\epsilon_j(t) = \int_0^t ds\left(\frac{\gamma_{k;j}^{\mathbf{w}_+}(s)}{\mathbf{w}_{+;j}(s)} + \frac{\gamma_{k;j}^{\mathbf{w}_-}(s)}{\mathbf{w}_{-;j}(s)}\right), \qquad (80)$$

*then we have*

$$\kappa_j(t) = \kappa_j(0)e^{-\eta\epsilon_j(t)}. \qquad (81)$$

*Proof.* The proof applies the dynamics of $\mathbf{w}_+$ and that of $\mathbf{w}_-$:

$$
\begin{aligned}
\frac{d\kappa}{dt} &= \dot{\mathbf{w}}_+ \odot \mathbf{w}_- + \dot{\mathbf{w}}_- \odot \mathbf{w}_+ \\
&= \left( -\frac{\nabla_{\mathbf{w}_+} L}{1-\mu} - \eta \gamma_k^{\mathbf{w}_+} \right) \odot \mathbf{w}_- + \mathbf{w}_+ \odot \left( -\frac{\nabla_{\mathbf{w}_-} L}{1-\mu} - \eta \gamma_k^{\mathbf{w}_-} \right) \\
&= -\eta \left( \gamma_k^{\mathbf{w}_+} \odot \mathbf{w}_- + \mathbf{w}_+ \odot \gamma_k^{\mathbf{w}_-} \right),
\end{aligned}
\tag{82}
$$

where we use Eq. (26) in the second equality and Eq. (79) in the third equality. As a result, for the $j$-th component of $\kappa$, we have

$$
\dot{\kappa}_j = -\eta \kappa_j \left( \frac{\gamma_{k;j}^{\mathbf{w}_+}}{\mathbf{w}_{+;j}} + \frac{\gamma_{k;j}^{\mathbf{w}_-}}{\mathbf{w}_{-;j}} \right)
$$

$$
\implies \kappa_j(t) = \kappa_j(0) e^{-\eta \epsilon_j(t)}.
\tag{83}
$$

$\square$

It is also interesting to investigate the dynamics of $\mathbf{w}$ as shown below.

**Lemma B.2.** *If $\mathbf{w}_\pm$ is run with HBF, then the dynamics of $\mathbf{w}$ satisfies that*

$$
\dot{\mathbf{w}} = -4\mathbf{v} \odot \frac{\nabla_{\mathbf{w}} L}{1-\mu} - \eta \Gamma_k^{\mathbf{w}}
\tag{84}
$$

*where we let*

$$
\mathbf{v} = (\mathbf{w}_+ \odot \mathbf{w}_+ + \mathbf{w}_- \odot \mathbf{w}_-), \quad \Gamma_k^{\mathbf{w}} = 2 \left( \gamma_k^{\mathbf{w}_+} \odot \mathbf{w}_+ - \gamma_k^{\mathbf{w}_-} \odot \mathbf{w}_- \right).
\tag{85}
$$

*Proof.* Using the dynamics of $\mathbf{w}_\pm$ Eq. (26), we can show that

$$
\begin{aligned}
\dot{\mathbf{w}} &= 2\dot{\mathbf{w}}_+ \odot \mathbf{w}_+ - 2\dot{\mathbf{w}}_- \odot \mathbf{w}_- \\
&= 2 \left( -\frac{\nabla_{\mathbf{w}_+} L}{1-\mu} - \eta \gamma_k^{\mathbf{w}_+} \right) \odot \mathbf{w}_+ - 2 \left( -\frac{\nabla_{\mathbf{w}_-} L}{1-\mu} - \eta \gamma_k^{\mathbf{w}_-} \right) \odot \mathbf{w}_- \\
&= -4 \left( \mathbf{w}_+ \odot \mathbf{w}_+ + \mathbf{w}_- \odot \mathbf{w}_- \right) \odot \frac{\nabla_{\mathbf{w}} L}{1-\mu} - 2\eta \left( \gamma_k^{\mathbf{w}_+} \odot \mathbf{w}_+ - \gamma_k^{\mathbf{w}_-} \odot \mathbf{w}_- \right).
\end{aligned}
\tag{86}
$$

$\square$

To show the implicit bias of HBF, we need to first explore the dynamics of $\mathbf{w}$, which is present in the following lemma.

**Lemma B.3** (Dynamics of $\mathbf{w}$ for diagonal linear networks under HBF)**.** *Under conditions of Theorem 3.1, if the diagonal linear network $f(x; \mathbf{w})$ is trained with HBF (Theorem 2.1), let*

$$
\Lambda_j^{\mathrm{GF}}(\mathbf{w}; \kappa(t)) = \frac{2\kappa_j(t)}{4} \left[ \frac{\mathbf{w}_j(t)}{2\kappa_j(t)} \operatorname{arcsinh}\left( \frac{\mathbf{w}_j(t)}{2\kappa_j(t)} \right) - \sqrt{1 + \frac{\mathbf{w}_j^2(t)}{4\kappa_j^2(t)}} + 1 \right]
$$

$$
\varphi_j(t) = \frac{\eta}{4} \int_0^t ds \left[ \frac{\gamma_{k;j}^{\mathbf{w}_+}(s)}{\mathbf{w}_{+;j}(s)} - \frac{\gamma_{k;j}^{\mathbf{w}_-}(s)}{\mathbf{w}_{-;j}(s)} \right]
$$

$$
\Lambda_j(\mathbf{w}, t; \kappa) = \Lambda_j^{\mathrm{GF}}(\mathbf{w}; \kappa(t)) + \mathbf{w}_j(t) \varphi_j(t),
\tag{87}
$$

*then the learning dynamics of the parameter $\mathbf{w}$ satisfies that*

$$
\frac{d}{dt} \partial_{\mathbf{w}_j} \Lambda_j + \frac{\partial_{\mathbf{w}_j} L}{1-\mu} = 0.
\tag{88}
$$

The proof of this lemma can be found in Appendix B.2. In the following we first focus on the proof of Theorem 3.1.

## B.1 Proof of Theorem 3.1

Now we can prove Theorem 3.1 with above helper lemmas.

*Proof.* Recall the definition of $\Lambda_j$ in Lemma B.3 and we further define

$$\Lambda(\mathbf{w}, t; \kappa) = \sum_{j=1}^{d} \Lambda_j(\mathbf{w}, t; \kappa), \tag{89}$$

then Lemma B.3 gives us

$$\frac{d}{dt} \nabla_{\mathbf{w}} \Lambda(\mathbf{w}, t; \kappa) = \left( \frac{d}{dt} \partial_{\mathbf{w}_1} \Lambda_1(\mathbf{w}, t; \kappa), \dots, \frac{d}{dt} \partial_{\mathbf{w}_d} \Lambda_d(\mathbf{w}, t; \kappa) \right)^T$$

$$= -\frac{X^T r}{n(1 - \mu)}$$

$$\implies \nabla_{\mathbf{w}} \Lambda(\mathbf{w}(\infty), \infty; \kappa(\infty)) - \nabla_{\mathbf{w}} \Lambda(\mathbf{w}(0), 0; \kappa(0)) = -\sum_{i=1}^{n} \frac{x_i \int_0^\infty r_i(\tau) d\tau}{n(1 - \mu)} = \sum_{i=1}^{n} x_i c_i \tag{90}$$

where we let $c_i = -\frac{\int_0^\infty r_i(\tau) d\tau}{n(1-\mu)}$. Let $\nabla_{\mathbf{w}} \Lambda(\mathbf{w}(0), 0; \kappa(0)) = 0$ and recall the definition of $\Lambda(\mathbf{w}; \kappa)$ in Theorem 3.1., then Eq. (90) is equivalent to

$$\nabla_{\mathbf{w}} \Lambda(\mathbf{w}; \kappa) - \sum_{i=1}^{n} x_i c_i = 0,$$

which is exactly the KKT condition of $\arg\min_{\mathbf{w}: X\mathbf{w}=y} \Lambda(\mathbf{w}; \kappa)$ proposed in Theorem 3.1. Therefore, we finish the proof. $\qquad\square$

## B.2 Proof of Lemma B.3

In this section we present the proof of Lemma B.3.

*Proof.* For simplicity, in the following we write the subscripts explicitly. According to Lemma B.2, the dynamics of $\mathbf{w}_j$ can be written as

$$\dot{\mathbf{w}}_j = -\frac{4}{1 - \mu} \mathbf{v}_j \partial_{\mathbf{w}_j} L - \eta \Gamma_{k;j}^{\mathbf{w}}. \tag{91}$$

Note that

$$\mathbf{v}_j^2 - \mathbf{w}_j^2 = 4\mathbf{w}_{+;j}^2 \mathbf{w}_{-;j}^2 \implies \mathbf{v}_j^2 = \sqrt{\mathbf{w}_j^2 + 4\kappa_j^2}, \tag{92}$$

then Eq. (91) can be written as

$$\frac{\dot{\mathbf{w}}_j}{4\sqrt{\mathbf{w}_j^2 + 4\kappa_j^2}} = -\frac{\partial_{\mathbf{w}_j} L}{1 - \mu} - \eta \frac{\Gamma_{k;j}^{\mathbf{w}}}{4\sqrt{\mathbf{w}_j^2 + 4\kappa_j^2}}. \tag{93}$$

We now define a function

$$\Lambda_j(\mathbf{w}, t; \kappa) = \bar{\Lambda}_j(\mathbf{w}, t; \kappa) + \mathbf{w}_j \varphi_j(t) \tag{94}$$

for some $\bar{\Lambda}_j(\mathbf{w}, t; \kappa)$ and $\varphi_j(t)$ such that

$$\frac{d}{dt} \partial_{\mathbf{w}_j} \Lambda_j(\mathbf{w}, t; \kappa) = \frac{\dot{\mathbf{w}}_j + \eta \Gamma_{k;j}^{\mathbf{w}}}{4\sqrt{\mathbf{w}_j^2 + 4\kappa_j^2}}, \tag{95}$$

the we can prove this lemma. Now we continue to find the $\bar{\Lambda}_j(\mathbf{w}, t; \kappa)$ and $\varphi_j(t)$. By definition,

$$\frac{d}{dt} \partial_{\mathbf{w}_j} \Lambda_j(\mathbf{w}, t; \kappa) = \partial_{\mathbf{w}_j}^2 \bar{\Lambda}_j \dot{\mathbf{w}}_j + \partial_t \partial_{\mathbf{w}_j} \bar{\Lambda}_j + \dot{\varphi}_j, \tag{96}$$

which, when compared with Eq. (95), implies that

$$\partial^2_{\mathbf{w}_j} \bar{\Lambda}_j = \frac{1}{4\sqrt{\mathbf{w}_j^2 + 4\kappa_j^2}}. \tag{97}$$

Solving this equation gives us

$$\partial_{\mathbf{w}_j} \bar{\Lambda}_j = \frac{1}{4} \int \frac{d\mathbf{w}_j}{\sqrt{\mathbf{w}_j^2 + 4\kappa_j^2}} = \frac{\ln\left(\sqrt{\mathbf{w}_j^2 + 4\kappa_j^2} + \mathbf{w}_j\right)}{4} + c \tag{98}$$

where $c$ is a constant and can be determined by requiring $\partial_{\mathbf{w}_j} \bar{\Lambda}_j|_{t=0} + \varphi_j(0) = 0 \implies c = -\ln(2\kappa_j(0))/4$. Thus Eq. (98) becomes

$$\partial_{\mathbf{w}_j} \bar{\Lambda}_j = \frac{1}{4} \ln\left(\frac{\sqrt{\mathbf{w}_j^2 + 4\kappa_j^2(t)} + \mathbf{w}_j}{2\kappa_j(t)}\right) - \frac{\eta\epsilon_j(t)}{4}$$

where we have used the definition of $\epsilon_j(t)$ in Lemma B.1. Solving the above equation will give us the form of $\bar{\Lambda}_j$

$$\begin{aligned}
\bar{\Lambda}_j &= \frac{1}{4} \int d\mathbf{w}_j \operatorname{arcsinh}\left(\frac{\mathbf{w}_j}{2\kappa_j(t)}\right) - \frac{\eta\epsilon_j(t)\mathbf{w}_j}{4} \\
&= \frac{1}{4}\left[\mathbf{w}_j \operatorname{arcsinh}\left(\frac{\mathbf{w}_j}{2\kappa_j(t)}\right) - \sqrt{\mathbf{w}_j^2 + 4\kappa_j^2(t)} + 2\kappa_j(t)\right] - \frac{\eta\epsilon_j(t)\mathbf{w}_j}{4} \\
&= \Lambda_j^{\mathrm{GF}}(\mathbf{w}; \kappa(t)) - \frac{\eta\epsilon_j(t)\mathbf{w}_j}{4} \tag{99}
\end{aligned}$$

where we use the definition of $\Lambda^{\mathrm{GF}}$ in Eq. (25). Comparing the rest parts of Eq. (96) with Eq. (95) requires that

$$\partial_t \partial_{\mathbf{w}_j} \bar{\Lambda}_j + \dot{\varphi}_j = \eta \frac{\Gamma^{\mathbf{w}}_{k;j}}{4\sqrt{\mathbf{w}_j^2 + 4\kappa_j^2(t)}}$$

$$\implies \dot{\varphi}_j(t) = \frac{\eta\kappa_j^2(t)\dot{\epsilon}_j(t)}{\left(\mathbf{w}_j + \sqrt{\mathbf{w}_j^2 + 4\kappa_j^2(t)}\right)\sqrt{\mathbf{w}_j^2 + 4\kappa_j^2(t)}} + \eta \frac{\Gamma^{\mathbf{w}}_{k;j}}{4\sqrt{\mathbf{w}_j^2 + 4\kappa_j^2(t)}}. \tag{100}$$

When combined with the form of $\bar{\Lambda}_j$, we can find the form of $\Lambda_j$:

$$\begin{aligned}
\Lambda_j(\mathbf{w}, t; \kappa) &= \Lambda_j^{\mathrm{GF}}(\mathbf{w}; \kappa(t)) + \eta\mathbf{w}_j \int \frac{ds}{\sqrt{\mathbf{w}_j^2 + 4\kappa_j^2(s)}}\left[\frac{\kappa_j^2(s)}{\mathbf{w}_j + \sqrt{\mathbf{w}_j^2 + 4\kappa_j^2(s)}}\dot{\epsilon}_j\right. \\
&\qquad\qquad \left. - \frac{\sqrt{\mathbf{w}_j^2 + 4\kappa_j^2(s)}\dot{\epsilon}_j}{4} + \frac{\Gamma^{\mathbf{w}}_{k;j}}{4}\right] \\
&= \Lambda_j^{\mathrm{GF}}(\mathbf{w}; \kappa(t)) + \eta\mathbf{w}_j \int \frac{ds}{\sqrt{\mathbf{w}_j^2 + 4\kappa_j^2(s)}}\left[-\frac{\mathbf{w}_j\dot{\epsilon}_j}{4} + \frac{\Gamma^{\mathbf{w}}_{k;j}}{4}\right] \\
&= \Lambda_j^{\mathrm{GF}}(\mathbf{w}; \kappa(t)) + \eta\mathbf{w}_j \int ds \left[\frac{\gamma^{\mathbf{w}+}_{k;j}}{\mathbf{w}_{+;j}} - \frac{\gamma^{\mathbf{w}-}_{k;j}}{\mathbf{w}_{-;j}}\right] \tag{101}
\end{aligned}$$

where we use the definition of $\epsilon_j$ (Lemma B.1) and Eq. (92) in the last equality. $\qquad\square$

## B.3 Implicit Bias of HBF for Diagonal Linear Networks when $\alpha = 2$

In this case, the correction term $\gamma^{\mathbf{w}\pm}$ will be

$$\gamma^{\mathbf{w}\pm} = \frac{1+\mu}{2(1-\mu)^3} \nabla_{\mathbf{w}_\pm} L \cdot \nabla^2_{\mathbf{w}_\pm} L.$$

We need to first find the Hessian $\nabla^2_{\mathbf{w}_\pm} L$. Due to the element-wise product, it will be convenient to derive the Hessian by writing the subscripts explicitly. We start with $\mathbf{w}_+$.

$$\partial_{\mathbf{w}_{+;i}}\partial_{\mathbf{w}_{+;j}} L = \frac{2}{n}\partial_{\mathbf{w}_{+;i}}\left(\mathbf{w}_{+;j}(X^T r)_j\right)$$

$$= \frac{2}{n}\left[\delta_{ij}(X^T r)_j + \sum_{c=1}^{n}\mathbf{w}_{+;j}\partial_{\mathbf{w}_{+;i}}\left(x_{c;j}(x_c^T\mathbf{w} - y_c)\right)\right]$$

$$= \frac{2}{n}\left[\delta_{ij}(X^T r)_j + 2\sum_{c=1}^{n}\mathbf{w}_{+;j}x_{c;j}x_{c;i}\mathbf{w}_{+;i}\right], \tag{102}$$

where we use the delta symbol $\delta_{ij} = 1$ if $i = j$ otherwise $\delta_{ij} = 0$. Therefore, we can conclude that

$$\nabla^2_{\mathbf{w}_+} L = \frac{2}{n}\left[\mathrm{diag}(X^T r) + 2\sum_{c=1}^{n}(\mathbf{w}_+ \odot x_c)(\mathbf{w}_+ \odot x_c)^T\right]. \tag{103}$$

Following a similar approach, we obtain that for $\mathbf{w}_-$

$$\partial_{\mathbf{w}_{-;i}}\partial_{\mathbf{w}_{-;j}} L = -\frac{2}{n}\partial_{\mathbf{w}_{-;i}}\left(\mathbf{w}_{-;j}(X^T r)_j\right)$$

$$= \frac{2}{n}\left[-\delta_{ij}(X^T r)_j + 2\sum_{c=1}^{n}\mathbf{w}_{-;j}x_{c;j}x_{c;i}\mathbf{w}_{-;i}\right] \tag{104}$$

$$\implies \nabla^2_{\mathbf{w}_-} L = \frac{2}{n}\left[-\mathrm{diag}(X^T r) + 2\sum_{c=1}^{n}(\mathbf{w}_- \odot x_c)(\mathbf{w}_- \odot x_c)^T\right]. \tag{105}$$

It is now left for us to find the form of $\nabla_{\mathbf{w}_\pm} L \cdot \nabla^2_{\mathbf{w}} L$. Again, it is convenient to write the subscripts explicitly:

$$\left(\nabla_{\mathbf{w}_+} L \cdot \nabla^2_{\mathbf{w}_+} L\right)_j = \sum_{i=1}^{d}\partial_{\mathbf{w}_{+;i}}\partial_{\mathbf{w}_{+;j}} L\partial_{\mathbf{w}_{+;i}} L$$

$$= \frac{4}{n^2}\sum_{i=1}^{d}\left[\delta_{ij}(X^T r)_j + 2\sum_{c=1}^{n}\mathbf{w}_{+;j}x_{c;j}x_{c;i}\mathbf{w}_{+;i}\right]\mathbf{w}_{+;i}(X^T r)_i$$

$$= \frac{4}{n^2}\left[\mathbf{w}_{+;j}((X^T r)_j)^2 + 2\sum_{c=1}^{n}\mathbf{w}_{+;j}x_{c;j}\left(x_c \odot \mathbf{w}_+ \odot \mathbf{w}_+\right)^T X^T r\right]. \tag{106}$$

Similarly,

$$\left(\nabla_{\mathbf{w}_-} L \cdot \nabla^2_{\mathbf{w}_-} L\right)_j = \frac{4}{n^2}\left[\mathbf{w}_{-;j}((X^T r)_j)^2 - 2\sum_{c=1}^{n}\mathbf{w}_{-;j}x_{c;j}\left(x_c \odot \mathbf{w}_- \odot \mathbf{w}_-\right)^T X^T r\right]. \tag{107}$$

Using Eq. (106) and (107), we can derive that

$$\frac{\gamma_j^{\mathbf{w}_\pm}}{\mathbf{w}_{\pm;j}} = \frac{2(1+\mu)}{(1-\mu)^3 n^2}\left[((X^T r)_j)^2 \pm 2\sum_{c=1}^{n}x_{c;j}\left(x_c \odot \mathbf{w}_\pm \odot \mathbf{w}_\pm\right)^T X^T r\right], \tag{108}$$

which further gives us the integral $\epsilon_j$:

$$\dot{\epsilon}_j = \frac{\gamma_j^{\mathbf{w}_+}}{\mathbf{w}_{+;j}} + \frac{\gamma_j^{\mathbf{w}_-}}{\mathbf{w}_{-;j}}$$

$$= \frac{4(1+\mu)}{(1-\mu)^3 n^2}\left[((X^T r)_j)^2 + \sum_{c=1}^{n}\sum_{i=1}^{d}x_{c;j}x_{c;i}(X^T r)_i\left(\mathbf{w}^2_{+;i} - \mathbf{w}^2_{-;i}\right)\right]$$

$$= \frac{4(1+\mu)}{(1-\mu)^3 n^2}\left[((X^T r)_j)^2 + \sum_{c=1}^{n}x_{c;j}x_c^T(\mathbf{w} \odot (X^T r))\right]$$

$$= \frac{4(1+\mu)}{(1-\mu)^3}\left[(\nabla_{\mathbf{w}} L)_j^2 + \frac{1}{n}\left(X^T X(\mathbf{w} \odot \nabla_{\mathbf{w}} L)\right)_j\right]. \tag{109}$$

On the other hand, according to Lemma B.2, $\partial_{\mathbf{w}_i} L$ can be written as

$$-(1-\mu)\frac{\dot{\mathbf{w}}_i}{4\mathbf{v}_i} - \eta(1-\mu)\frac{\Gamma_i^{\mathbf{w}}}{4\mathbf{v}_i}, \tag{110}$$

which further gives us that

$$\eta \int_0^t ds \left(X^T X(\mathbf{w} \odot \nabla_{\mathbf{w}} L)\right)_j = -\eta(1-\mu)\sum_{c=1}^n\sum_{i=1}^d x_{c;j}x_{c;i}\int_{\mathbf{w}_i(0)}^{\mathbf{w}_i(t)} d\mathbf{w}_i \frac{\mathbf{w}_i(s)}{4\mathbf{v}_i(s)} + \mathcal{O}\left(\eta^2\right)$$

$$= -\eta(1-\mu)\sum_{c=1}^n\sum_{i=1}^d x_{c;j}x_{c;i}\int_{\mathbf{w}_i(0)}^{\mathbf{w}_i(t)} d\mathbf{w}_i \frac{\mathbf{w}_i(s)}{4\sqrt{\mathbf{w}_i^2(s)+4\kappa_i^2(s)}} + \mathcal{O}\left(\eta^2\right)$$

$$= -\frac{\eta(1-\mu)}{4}\sum_{c=1}^n\sum_{i=1}^d x_{c;j}x_{c;i}\left(\sqrt{\mathbf{w}_i^2(t)+4\kappa_i^2(t)} - \sqrt{\mathbf{w}_i^2(0)+4\kappa_i^2(0)}\right).$$

where we use Lemma B.2 in the first equality and Eq. (92) in the second equality. Since $\mathbf{w}(0)=0$ and Lemma B.1, we obtain

$$\eta \int_0^t ds \left(X^T X(\mathbf{w}\odot\nabla_{\mathbf{w}} L)\right)_j = -\frac{\eta(1-\mu)}{4}\sum_{c=1}^n\sum_{i=1}^d x_{c;j}x_{c;i}\left(\sqrt{\mathbf{w}_i^2(t)+4\kappa_i^2(0)} - 2\kappa_i(0)\right)$$

$$= -\frac{\eta(1-\mu)}{4}\left(X^T X\mathbf{q}(t)\right)_j \tag{111}$$

where we let $\mathbf{q}\in\mathbb{R}^d$ with

$$\mathbf{q}_i(t) = \sqrt{\mathbf{w}_i^2(t)+4\kappa_i^2(0)} - 2\kappa_i(0) \geq 0.$$

Now combining Eq. (109) and Eq. (111), we can derive

$$\eta\epsilon_j(t) = \frac{4\eta(1+\mu)}{(1-\mu)^3}\int_0^t ds(\partial_{\mathbf{w}_j} L)^2 - \frac{\eta(1+\mu)}{(1-\mu)^2 n}\left(X^T X\mathbf{q}\right)_j + \mathcal{O}\left(\eta^2\right). \tag{112}$$

To obtain the full potential function, we still need to find the form of $\varphi_j$. According to the definition of $\mathbf{v}$ and $\epsilon_j$ and Eq. (108), we can derive

$$2\gamma_{k;j}^{\mathbf{w}_+}\mathbf{w}_{+;j} - 2\gamma_{k;j}^{\mathbf{w}_-}\mathbf{w}_{-;j} - \mathbf{w}_j\dot{\epsilon}_j = \mathbf{v}_j\left(\frac{\gamma_{k;j}^{\mathbf{w}_+}}{\mathbf{w}_+} - \frac{\gamma_{k;j}^{\mathbf{w}_-}}{\mathbf{w}_-}\right)$$

$$= \frac{4(1+\mu)}{(1-\mu)^3 n}\sum_{c=1}^n\sum_{i=1}^d \mathbf{v}_j x_{c;j}x_{c;i}\mathbf{v}_i\partial_{\mathbf{w}_i} L, \tag{113}$$

which, when combined with the definition of $\varphi_j$ in Lemma B.3, further gives us

$$\dot{\varphi}_j = \eta\frac{(1+\mu)}{(1-\mu)^3 n}\sum_{c=1}^n\sum_{i=1}^d x_{c;j}x_{c;i}\mathbf{v}_i\partial_{\mathbf{w}_i} L$$

$$= -\frac{\eta(1+\mu)}{4(1-\mu)^2 n}\sum_{c=1}^n\sum_{i=1}^d x_{c;j}x_{c;i}\dot{\mathbf{w}}_i + \mathcal{O}\left(\eta^2\right) \tag{114}$$

where we use Eq. (110) in the second equality. As a result,

$$\varphi_j(\infty) = -\frac{\eta(1+\mu)}{4(1-\mu)^2 n}\sum_{c=1}^n\sum_{i=1}^d x_{c;j}x_{c;i}\mathbf{w}_i(t) = \frac{\eta(1+\mu)}{4(1-\mu)^2 n}\left(X^T X\mathbf{w}\right)_j. \tag{115}$$

One interesting thing aspect of $\varphi_j$ if $\mathbf{w}$ converges to an interpolation solution where $X\mathbf{w}(\infty)=y$ is

$$\varphi_j(\infty) = \frac{\eta(1+\mu)}{4(1-\mu)^2}\partial_{\mathbf{w}_j} L(0). \tag{116}$$

In summary, the potential function $\mathcal{O}\left(\eta^2\right)$-close HBF is

$$\kappa_j(\infty) = \kappa_j(0)\exp\left(-\frac{4\eta(1+\mu)}{(1-\mu)^3}\int_0^\infty ds(\partial_{\mathbf{w}_j} L)^2 + \frac{\eta(1+\mu)}{(1-\mu)^2 n}\left(X^T X\mathbf{q}(\infty)\right)_j\right)$$

$$\Lambda_j(\mathbf{w},\infty;\kappa) = \Lambda_j^{\mathrm{GF}}(\mathbf{w},\kappa(\infty)) + \frac{\eta(1+\mu)}{4(1-\mu)^2}\mathbf{w}_j\partial_{\mathbf{w}_j} L(0). \tag{117}$$

## C  Details for Numerical experiments

The experiments are conducted on a CentOS Linux 7.9.2 platform equipped with an Intel(R) Xeon CPU E5-2683 at 3.00 GHz, 256GB of RAM, and an NVIDIA Tesla A100 graphics card.

### C.1  Details for Section 2.2.2

For the experiment of Figure 1, we conduct observation on the comparison of directional smoothness for HB and GD on the CIFAR-10 dataset Krizhevsky et al. (2009). A multilayer perceptron with two hidden layers (each of width 200) is trained for 2000 epochs using full-batch GD and HB ($\mu = 0.9$), and the step size is set to 0.1.

### C.2  Details for Section 4

For the discrete learning dynamics of HB and GD, we set the step size as $\eta$ and the momentum factor is $\mu$. For the continuous approximations, we use $\eta_{\text{Euler}} = \eta/10$ as the Euler step sizes to approximate the dynamics. These hyper-parameters are listed in Table 2.

| $x, y$ | $1, 0.6$ |
|---|---|
| Starting point | $a_1 = 2.8, a_2 = 3.5$ |
| $\eta$ | $5 \times 10^{-3}$ |
| $\mu$ | $0.7$ |
| $\eta_{\text{Euler}}$ | $5 \times 10^{-4}$ |

Table 2: Hyper-parameters for 2-d model.

We let the model parameter be $\beta = (a_1, a_2)^T \in \mathbb{R}^2$. For RGF, we use the ODE

$$\dot{\beta} = -\frac{\nabla_\beta L}{1 - \mu} \implies \beta_{k+1} = \beta_k - \eta_{\text{Euler}} \frac{\nabla_\beta L}{1 - \mu}.$$

Formulations of HBFs with $\alpha = 2, 3$ are denoted in Table 1. We denote $\mathbf{1}_d = (1, \ldots, 1)^T \in \mathbb{R}^d$. For the dataset $\{(x_i, y_i)\}_{i=1}^d$, we set $n = 40, d = 100$. The data point follows a Gaussian distribution $\mathcal{N}(0, I_d)$. To make the ground truth solution $\mathbf{w}^*$ sparse, we let 5 components of it be nonzero. Recall that the initialization is $\kappa(0) = s^2 \mathbf{1}_d$ where $s$ controls the initialization scale. In Fig. 6(b), we make the initialization as $\mathbf{w}_+ = \mathbf{w}_- = s\mathbf{1}_d$ with $s = 0.01$. We set the step size $\eta$ for HB as $10^{-3}$. For RGF and HBF, we let the Euler step size $\eta_{\text{Euler}} = 10^{-4}$ to simulate the continuous dynamics. In Fig. 6(a) and 6(b), we set $\eta = 10^{-2}$. For the initialization, to make the task slightly harder, we let $\mathbf{w}_+ = \vartheta s \mathbf{1}_d$ and $\mathbf{w}_- = s\mathbf{1}_d/\vartheta$ with $\vartheta = 0.9$ such that we still have $\kappa(0) = s^2 \mathbf{1}_d$ while the initialization symmetry is slightly broken.

**Addition experiments for different $\eta_{\text{Euler}}$**  We additionally run the experiments in Fig. 2 for each of $\eta_{\text{Euler}} = \{\eta/10, \eta/100, \eta/1000\}$, and confirm that the observations and conclusions hold in all different $\eta_{\text{Euler}}$ Fig. 3.

### C.3  Additional Numerical Experiments for Non-Linear Networks

We conduct experiments of Fig. 2(b) in the MNIST dataset, where we now train a three-layer fully-connected neural networks (FCNN). The FCNN has a structure of `Linear(784×128)` $\rightarrow$ `SiLU`$\rightarrow$`Linear(128×128)`$\rightarrow$`BatchNormalization`$\rightarrow$`Linear(128×10)`. Cross-entropy is used for the loss function. The batch size is 60,000 and the momentum factor $\mu \in (0.7, 0.8)$. The learning rate is $\eta = 0.01$. The results (reported in Fig. 4) well align with the results for the toy model in Fig. 2(b): HBF with $\alpha = 3$ has lower discretization error compared to HBF with $\alpha = 2$, and both of them are better than the RGF.

### C.4  Additional Numerical Experiments for Diagonal Linear Networks

We now investigate the implicit bias of HB for 2-layer diagonal linear networks

$$f(x; \mathbf{w}) = \mathbf{w}^T x = (\mathbf{w}_+ \odot \mathbf{w}_+ - \mathbf{w}_- \odot \mathbf{w}_-)^T x \tag{118}$$

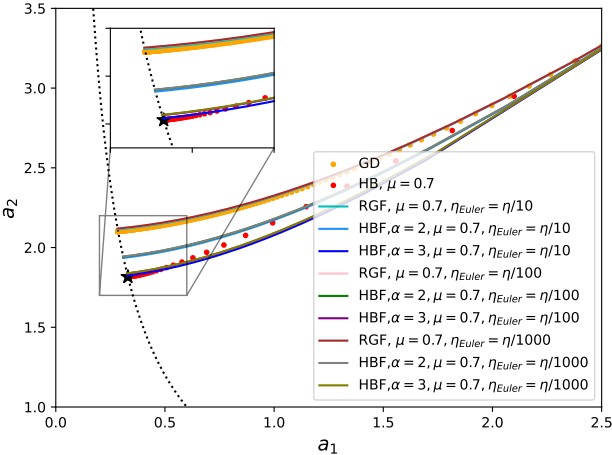

Figure 3: Trajectories of continuous approximations with different $\eta_{\text{Euler}}$.

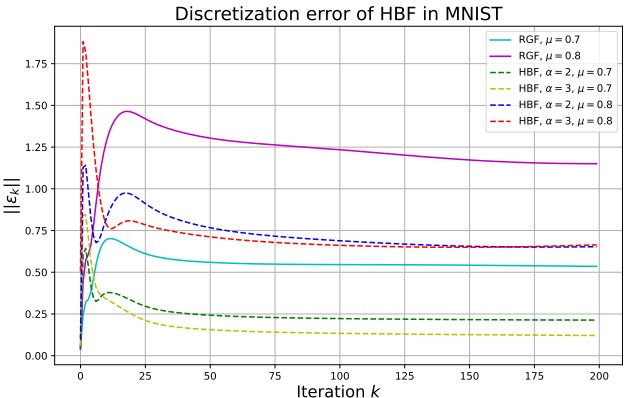

Figure 4: Discretization error of HBF in MNIST when training a three non-linear networks.

for a dataset $\{(x_i, y_i)\}_{i=1}^n$ where $x \in \mathbb{R}^d, y \in \mathbb{R}$. The empirical loss is

$$L(\mathbf{w}_+, \mathbf{w}_-) = \sum_i (f(x_i; \mathbf{w}) - y_i)^2. \tag{119}$$

We let $n < d$ and denote the ground truth solution as $\mathbf{w}^*$ such that $\mathbf{w}^{*T} x = y$. We let $\mathbf{w}^*$ be sparse. For a given scale $s$ we let

$$\kappa(0) = \mathbf{w}_+(0) \odot \mathbf{w}_-(0) = s^2 (1, \ldots, 1)^T \in \mathbb{R}^d. \tag{120}$$

### C.4.1 Discretization Error for Different Approximations

Our first experiment explores the discretization error, where we let $k$ denote the iteration count and first obtain $\mathbf{w}_k^{\text{HB}}$ by training $f(x; \mathbf{w})$ with HB. In addition, we also train $f(x; \mathbf{w})$ with RGF (Eq. (1)) and HBF (Corollary 3.2), respectively. We calculate the discretization error as

$$\|\mathbf{w}_k^{\text{HB}} - \mathbf{w}(t_k)\|_2^2 \tag{121}$$

for $\mathbf{w}(t_k)$ obtained from HBF or RGF and present the results in Fig. 5, where HBF enjoys smaller discretization error than RGF for different $\mu$, supporting our theoretical claims.

### C.4.2 Implication for difference of implicit bias between HB and GD

We first discuss the implication obtained from our HBF. Setting $\mu = 0$ in Corollary 3.2 gives us the implicit bias of $\mathcal{O}(\eta^2)$-close continuous approximation of GD, i.e., IGR (Barrett and Dherin, 2022)

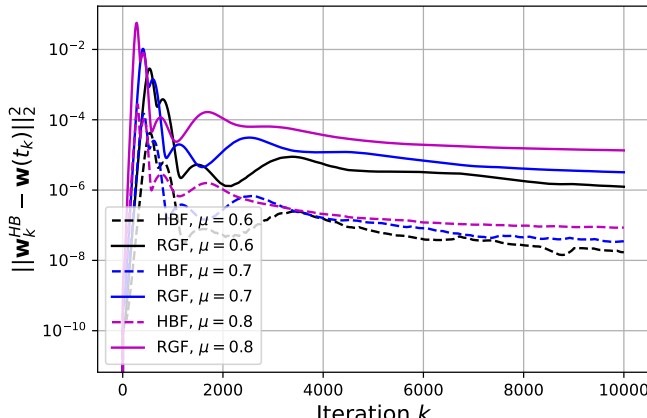

Figure 5: Discretization errors $\|\mathbf{w}_k^{\text{HB}} - \mathbf{w}(t_k)\|_2^2$ for HBF (dotted lines) and RGF (solid lines), respectively, when training the 2-layer diagonal linear networks.

Flow (IGRF). Both IGRF and HBF have the initialization rescaling effect. The difference between them is closely connected with the composed parameter

$$\psi := \frac{\eta(1+\mu)}{(1-\mu)^2} \tag{122}$$

and the value of $\Phi/(1-\mu)$. And the discrepancy between the implicit bias of HB and that of GD will be more obvious for large value of $\mu$. These observations stand in contrast to the case for $\mathcal{O}(\eta)$-close RGF, which cannot distinguish the implicit bias of HB from that of GD.

In addition, we note the following observations: (i). the value of

$$\kappa \propto \exp(-\frac{\eta 4(1+\mu)}{(1-\mu)^3} \int_0^\infty ds(\partial L)^2) \tag{123}$$

is related to the speed of convergence since it depends on the integral of gradient along the training trajectory, implying that a faster speed of convergence would possibly lead to a smaller $\int_0^\infty ds(\partial L)^2$ which then leads to a larger $\kappa$; (ii). a smaller $\kappa$ implies a better sparsity and generalization performance for sparse regression, because the objective function in Corollary 3.2 will be closer to the $\ell_1$-norm.

Now let $\kappa^{HBF}$ and $\kappa^{IGRF}$ be $\kappa$ obtained from HBF and IGRF, respectively. As HB converges faster than GD in practice, then $\partial L$ becomes neglectable very quickly for HBF if it converges too fast, and, according to our observation (i) above, we conclude that $\kappa^{HBF} > \kappa^{IGRF}$. Then according to our observation (ii) above, IGRF will generalize better than HBF. On the other hand, if the speed of convergence of HBF and that of IGRF are similar (e.g., blue line in Fig. 6(b)), then HBF and IGRF will have similar values of $\Phi = \int_0^\infty ds(\partial L)^2$ while $\kappa^{HBF}$ additionally depends on a coefficient $\frac{1+\mu}{(1-\mu)^3} > 1$, thus it is possible that in this cae $\kappa^{HBF} < \kappa^{IGRF}$, which implies that HBF will generalize better in this case. In summary, there might exist a tension between the speed of convergence and the generalization for HB, i.e., if HB converges too fast, $\kappa$ would be larger for HB hence solutions of GD would enjoy better generalization properties.

Below we conduct experiments for the above claims. In particular, we compare the implicit bias of HB with that of GD. Given $s$, we train $f(x; \mathbf{w})$ with GD and HB, respectively. We calculate the distance between the returned solution $\mathbf{w}(\infty)$ and the ground truth solution $\mathbf{w}^*$, i.e., $\|\mathbf{w}(\infty) - \mathbf{w}^*\|_2$, as a measure of generalization performance and report the results in Fig. 6(a). It can be seen that, when the initialization scale $s$ is small, solutions of GD generalize better than those of HB. This can be explained by Corollary 3.2: compared to GD, when $s$ is small, $L(\mathbf{w})$ decreases much faster for HB (green lines in Fig. 6(b)), which leads to a smaller $\int ds L(\mathbf{w})$ and weaker initialization mitigation effect, thus the solutions of HB generalize worse than GD solutions. Recall that in Corollary 3.2, as $\kappa_j(0) = s^2$ increases, $\Phi$ determines the generalization performances for HB and GD since it controls the extent of the initialization mitigation effect. Furthermore, $L(\mathbf{w})$ does not decrease much faster

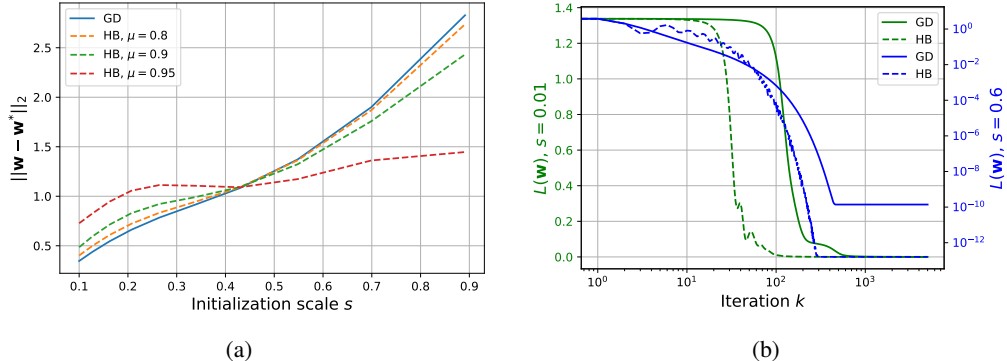

(a)                                              (b)

Figure 6: (a). Generalization performances $\|\mathbf{w}(\infty) - \mathbf{w}^*\|_2$ for different initialization scales $s$ when $f(x; \mathbf{w})$ is trained by GD and HB with different values of $\mu$. (b). $L(\mathbf{w})$ during training processes of HB ($\mu = 0.9$) and GD for different $s$.

for HB than for GD (blue lines in Fig. 6(b)), thus GD and HB have a similar value of $\Phi$, which is further enhanced by a factor of $(1 + \mu)/(1 - \mu)^3$ for HB according to Corollary 3.2. As a result, HB solutions will generalize better than GD and the discrepancy between them is more significant for large $\mu$ (large $(1 + \mu)/(1 - \mu)^3$) as shown in Fig. 6(a).

