# OpenReview forum: "Heavy-Ball Momentum Method in Continuous Time and Discretization Error Analysis"
_NeurIPS.cc/2025/Conference — NeurIPS 2025 poster_

### Official Review · Reviewer_iEY4 · 2025-06-06

**Clarity:** 3
**Significance:** 3
**Originality:** 2
**Rating:** 5
**Confidence:** 4

**Summary:**

The paper highlights the differences between the discrete and continuous time heavy-ball (HB) momentum methods.
The authors explore the continuous-time HB method and report notable findings that distinguish their work from previous studies.

- Research Question

    - The discretization error between gradient flow (GF) and gradient descent (GD) has been studied in [Barrett and Dherin, 2022; Miyagawa, 2023; Rosca et al., 2023]. Rather than GD, the authors focus on the HB method, which has yet to be explored.

    - More specifically, the continuous time HB as a surrogate model of the discrete HB has been formalized as a rescaled gradient flow (RGF: Eq. (1)) [Kovachki and Stuart (2020)], but the authors point out that it cannot differentiate gradient descent (GD) and HB (Sections 3 and 4). Another continuous time HB has been proposed by [Ghosh et al. (2023)], which reduces the discretization error compared to RGF, but it is limited to the second order of the step size. [Cattaneo et al. (2023)] has analyzed Momentum and Adam but is also limited to the second order.

- Solution

    - To address this issue, the authors propose HB Flow (HBF), a first-order piece-wise continuous differential equation built upon RGF, that approximates the discrete HB momentum method and can control discretization error to *arbitrary order* of the step size (learning rate).

    - The idea is to add a number of counter terms to RGF order-by-order (Theorem 2.1), akin to backward error analysis [Hairer et a., 2006; Miyagawa, 2023; Rosca et al., 2023].

- Applications

    - Examining the directional smoothness and the diagonal linear network, the authors demonstrate that HBF captures the differences between HB and GD, which has been impossible by RGF.

        - The authors show that the learning dynamics of HB exhibits smaller directional smoothness compared to GD (Fig. 1), through the lens of HBF.

        - In addition, they find the implicit bias of HB for the diagonal linear network (Theorem 3.1) that cannot be obtained by RGF.


- Overall, the paper argues that, despite its popularity, continuous-time analysis fails to fully capture the learning dynamics of discrete-time GD methods when neglecting discretization error. Although this limitation has been noted in prior work, such as [Miyagawa, 2023], it is worth emphasizing that the issue is general and extends to various optimization methods such as the HB momentum method.

**Questions:**

- When does the approximation accuracy of HBF for the discrete HB method deteriorate?

    - I suspect it occurs in regions where the derivatives of $L$ are large and where $1 - \mu \ll \eta$, causing the Taylor series to diverge.

    - Does this happen in practical settings?

    - Could you provide rough bounds on the derivatives required to keep the discretization error small?

    - It would also be helpful to provide a rough estimate of the $\alpha$\-th order discretization error of a $d$\-dimensional DNN at the $k$\-th step from the results in Section 2.2, for example.

- (Eq. (17) & (18)) This question relates to the one above. When $\mu \sim 1$, it appears that the dynamics are dominated by higher-order derivatives or even a constant term, rather than the first-order gradient. I am concerned that this does not reflect the correct learning dynamics. Why is that the case?

- How can we extend the authors’ approach to adaptive learning rate, e.g., Adam? What is the challenge specifically? The authors mentioned the extension to Adam in the Limitation section, and I agree this is an interesting and important step toward practically relevant scenarios in view of the prevalence of Adam.

    - To my knowledge, existing continuous-time formulations of Adam are quite complex. How might one begin to approach this problem?

- (Theorem 2.1) Can the result be extended from $\alpha \geq 1$ to $\alpha > 0$?

- To my understanding, a difference between the paper and [Miyagawa, 2023] is, aside from HB and GF, that the former proposes a piecewise continuous differential equation, whereas the latter provides a globally continuous one. Is it correct?

- (Appendix C.2) Was $\eta_\mathrm{Euler} = \eta/10$ sufficiently small? The authors can check this, for example, by changing $\eta/10, \eta/100,... $ and confirming the learning trajectory does not change dramatically or the change is much smaller than the discretization error, the quantity is of interest.

- How long did it take to run the experiments?

**Ethical Concerns:**

["NO or VERY MINOR ethics concerns only"]

**Final Justification:**

The paper presents strong technical contributions, offers novel insights, and is clearly written.

> - (Novelty) The authors explore the continuous-time heavy-ball (HB) method and report notable findings that distinguish their work from previous studies.
> - (Novelty & Technical Contribution) The authors show that the learning dynamics of HB exhibits smaller directional smoothness compared to GD (Fig. 1), through the lens of HBF.
> - (Novelty & Technical Contribution) They find the implicit bias of HB for the diagonal linear network (Theorem 3.1) that cannot be obtained by RGF.
> - (Broader Impact) The paper argues that, despite its popularity, continuous-time analysis fails to fully capture the learning dynamics of discrete-time GD methods when neglecting discretization error. Although this limitation has been noted in prior work, such as [Miyagawa, 2023], it is worth emphasizing that the issue is general and extends to various optimization methods such as the HB momentum method.

Specifically,
> - (Clarity, Novelty, & Technical Contribution) The contribution is clearly stated and significant for the field of optimization, as it helps bridge the gap between conventional continuous-time theory and practical discrete-time methods.
> - (Technical Contribution) The derivation of HBF is nontrivial$\textemdash$performing all-order backward error analysis involves careful calculations, including the use of Lie derivatives$\textemdash$demonstrating the paper's technical soundness.
> - (Novelty & Technical Contribution) All analyses in Section 3 and 4 are built upon the proposed HBF and thus are novel, indicating HBF can distinguish GD and HB unlike RGF.
> - (Soundness) The newly-observed implicit regularization of HBF discussed at the end of Section 2 matches our intuition: HB prefers smaller directional smoothness to GD because of the momentum (inertia).
> - (Empirical Validation) The experiment in Figure 2 is convincing: larger $\alpha$ leads to smaller discretization error. Although the model is small, this choice is acceptable given that computing higher-order derivatives becomes prohibitively expensive for large models.

In the rebuttal, most of the reviewers’ concerns have been addressed, and no critical issues remain.
The authors have provided additional experimental results on the dynamics of discretization error with a larger model and additional theoretical analysis on the bound of discretization error. These results have made the paper further convincing, thereby highlighting the contributions.
Moreover, the proof in the paper has been improved in the rebuttal, particularly regarding the $\mathcal{O}$ notation.

Therefore, I support the acceptance of this paper.

**Limitations:**

Yes, but additional discussions can be added: under what conditions the proposed HBF fails to serve as a valid and accurate approximation of the discrete-time HB method?

**Paper Formatting Concerns:**

No.

**Quality:**

4

**Strengths And Weaknesses:**

### Strengths

Please see also the Summary field:
> - (Novelty) The authors explore the continuous-time heavy-ball (HB) method and report notable findings that distinguish their work from previous studies.
> - (Novelty & Technical Contribution) The authors show that the learning dynamics of HB exhibits smaller directional smoothness compared to GD (Fig. 1), through the lens of HBF.
> - (Novelty & Technical Contribution) They find the implicit bias of HB for the diagonal linear network (Theorem 3.1) that cannot be obtained by RGF.
> - (Broader Impact) The paper argues that, despite its popularity, continuous-time analysis fails to fully capture the learning dynamics of discrete-time GD methods when neglecting discretization error. Although this limitation has been noted in prior work, such as [Miyagawa, 2023], it is worth emphasizing that the issue is general and extends to various optimization methods such as the HB momentum method.

- The proposed HBF might be seen as a natural progression of [Kovachki and Stuart (2020); Miyagawa, 2023; Rosca et al., 2023]. In fact, the the authors’ approach and contents in Section 2 in the paper are similar to  Sections 3 & 4 in [Miyagawa, 2023]. However, I do not think that this is a weakness because:

    - (Clarity, Novelty, & Technical Contribution) The contribution is clear$\textemdash$as summarized in the Summary field above$\textemdash$and significant for the field of optimization, as it helps bridge the gap between conventional continuous-time theory and practical discrete-time methods.

    - (Technical Contribution) In addition, the derivation of HBF is nontrivial$\textemdash$performing all-order backward error analysis involves careful calculations, including the use of Lie derivatives$\textemdash$demonstrating the paper's technical soundness.

    - (Novelty & Technical Contribution) All analyses in Section 3 and 4 are built upon the proposed HBF and thus are novel, indicating HBF can distinguish GD and HB unlike RGF.

- (Soundness) The newly-observed implicit regularization of HBF discussed at the end of Section 2 matches our intuition: HB prefers smaller directional smoothness to GD because of the momentum (inertia).

- (Empirical Validation) (If my concern below is addressed) The experiment in Figure 2 is convincing: larger $\alpha$ leads to smaller discretization error. Although the model is small, this choice is acceptable given that computing higher-order derivatives becomes prohibitively expensive for large models.


### Weaknesses

- (See Questions below) It would be helpful to clarify under what conditions the proposed HBF fails to serve as a valid and accurate approximation of the discrete time HB method.

- (See Questions below) I am concerned about $\eta_\mathrm{Euler}$.

- Additionally, I recommend that the authors provide their code for reproduction.


## Minor Comments

- [Comment (minor)] I recommend that the authors explicitly refer to $\gamma_k$ as the *counter term* in line 134. Currently, the term “counter term” appears only in the Abstract, Introduction, and Appendix A.1, which may lead to confusion for readers.

- [Comment (minor)] The colors in Figure 2(a) are hard to distinguish (three blues).

- [Typo] The numerator of $\frac{ }{q!}$ in Eq. (50) is missing.

- [Typo] (Lines 188, 193, 219, and 223) Theorem 3.1 should be Theorem 2.1?

- [Typo] (Line 831) Please Appendix C. -> Please see Appendix C.

- [Grammatical error] (Line 184) … it is worth to mention… -> … it is worth mentioning…

- [Comment (minor)] (Lines 192--194)

    > … the recursive manner in Theorem 3.1 provides a possibility to calculate the involved terms automatically by using software for symbolic mathematics such as SymPy, ….

    Totally agree $\textemdash$ combining SymPy with AI coding tools makes tedious analytical math way easier.


## Review Summary
The paper presents strong technical contributions, offers novel insights, and is clearly written. While it raises a few concerns noted above, I believe the strengths outweigh them. I recommend acceptance and look forward to the discussion with the authors.

---

> ### Author Rebuttal · Authors · 2025-07-31
>
> We thank the reviewer very much for the constructive suggestions and appreciation of our work. Below we answer your questions.
>
> ---
>
> ## Weaknesses Part
> 1. **Comment**: *"It would be...HB method."*
>
>    **Response**: We thank the reviewer for this insightful question. To make the HBF a valid approximation, there are two necessary conditions:
>     - It is crucial to control the ratio between $\eta $ and $1-\mu$ to avoid $\eta\gg1-\mu$, which might lead the Taylor series to diverge as suggested by the reviewer. More interestingly, we believe it is the magnitude of a special composite quantity $$\psi:=\frac{\eta}{(1-\mu)^2}\tag{a.1}$$that matters for the effectiveness of the HBF. This quantity spontaneously appears in both HBF with $\alpha=2$ and $\alpha=3$ but not in the RGF, i.e., for HBF with $\alpha=2$ the counter term is proportional to $\psi$ while for HBF with $\alpha=3$ it is proportional to $\psi^2$. If $\psi$ is too large, then our results would no longer hold. Hence, we need to fix the value of $\mu$ and treat only $\eta$ as the variable to denote the higher-order terms as $\mathcal{O}(\eta^{\alpha})$ while hide $\mu$ in the expansion. And it would be interesting for future works to study the case when both $\mu$ and $\eta$ are treated as variables such that higher-order terms can be denoted as $\mathcal{O}(\psi^{\alpha})$ (Eq.(a.1)) for $\alpha\geq1$.
>
>       In addition, the dependence of HBF on the special composite quantity $\psi$ is consistent with the empirical observation in Leclerc and Madry (2020), where the optimization curves for different momentum values can be recovered by a corresponding change in the learning rate. The dependence of HB on $\eta$ and $\mu$ at the same time further indicates the advantage of HBF with $\alpha > 1$ and that RGF, which only depends on $\mu$, is not sufficient to reflect the optimization properties of HB.
>     -  Given $\alpha$, the continuous approximations include derivatives of $L$ up to the $\alpha$-th order, hence $L$ should at least be $\alpha$-times continuously differentiable and $||\nabla^{\alpha}L||$ should be upper bounded.
> 2. **Comment**: *"I am concerned..."*
>
>    **Response**: We thank the reviewer for the suggestion. To validate the influence of the value of $\eta_{\text{euler}}$, we run the experiments in Fig.2 for each of $\eta_{\text{euler}}\in\\{\eta/10,\eta/100,\eta/1000\\}$, and confirm that the observations and conclusions hold in all different $\eta_{\text{euler}}$. As we are currently not allowed to upload any materials or links, we will add results of these new experiments in the revision.
> 3. **Comment**: *"Additionally, I...reproduction."*
>
>    **Response**: Thanks for the suggestion. We will upload the code, including the newly supplemented  numerical experiments, in the revision.
> ### Minor Comments
> - We thank the reviewer for this nice suggestion. We will formally define $\gamma_k$ as the counter term in the revision.
> - We have changed the colors so that these curves are now more discernible.
> - We thank the reviewer for pointing the typos and grammatical error, and we will correct them accordingly.
>
> ---
>
> ## Questions Part
> 1. **Comment**: *"When does the ... deteriorate?"*
>
>    **Response**: We thank the reviewer for the series of interesting questions. We answer these questions point by point below.
>    - We agree with the reviewer that the derivatives of $L$ are not large and that $\eta\ll1-\mu$ are two necessary conditions for HBF to be accurate approximations. Please see our response to the first point of the Weaknesses part.
>    - In practice, using large $\mu$ and small $\eta$ simultaneously can still reproduce the results of the case with small $\mu$ and large $\eta$, and both parameter sets correspond to a small $\psi$. This phenomenon has also been observed by Leclerc and Madry (2020).
>
>      However, the regime where both $\eta$ and $\mu$  are large (hence large $\psi$) will lead to failures of HBF due to the divergence of the counter terms, as suggested by the reviewer. In addition, in this regime, the model might not be trained properly either: the update direction coming from the gradient and that from the momentum will jointly affect the training direction significantly, while these two directions can be very different due to the large value of $\eta$ and $\mu$ hence cannot give a consistent updating direction.
>    - As the arbitrary order approximations involve complex combinations of higher-order derivatives, it would be intractable to derive an explicit bound. However, under the condition that the Taylor expansion is eligible, we can express the discretization error at the $k$-th step (please see below) with an explicit bound of the second-order derivative.
>    - We thank the reviewer for the nice suggestion. Below we give an estimate of the discretization error of the $\alpha$-th order continuous approximation at the $k$-th step. We first present several useful relations. As Eq.(43) is established by solving the functional equation for any iteration count $k$, we can write the relation between $\varepsilon_{k + 1}$ and $\varepsilon_{k}$ (Eq.(34), line 454-455 in page 13) as $$\varepsilon_{k+1}-\varepsilon_{k}=\mu(\varepsilon_{k}-\varepsilon_{k-1})-\eta [\nabla L(\beta(t_k)) - \nabla L(\beta(t_k) - \varepsilon_{k})]+\mathcal{O}(\eta^{\alpha+1}). \tag{a.2}$$If $\varepsilon_{k}=\mathcal{O}(\eta^{\alpha})$, then Eq.(a.2) implies $$||\varepsilon_{k+1}-\varepsilon_{k}||\leq\mu||\varepsilon_{k}-\varepsilon_{k-1}||+\eta\lambda||\varepsilon_{k}||+c_1\eta^{\alpha+1}$$for some constant $c_1$ where we use $G_k-\mu G_{k-1}=\nabla L$ and let $\lambda=\max_{\beta}||\nabla^2L(\beta)||$ be the upper bound of the second-order derivative. Denoting $$\forall k: c_2(k)=\frac{c_1}{\lambda}e^{\frac{2\lambda\eta}{1-\mu}k},\ c_3(k)=\frac{2c_1}{1-\mu}e^{\frac{2\lambda \eta}{1-\mu}k},$$we now prove by induction.
>      - For the first step ($k = 0$), by definition we have $$\varepsilon_0=0\leq c_2(0)\eta^{\alpha}.$$Note that $G_{-1}=0$ and $\gamma_{-1}=0$ by definition, then we have $$||\varepsilon_1-\varepsilon_0||\leq c_1\eta^{\alpha+1}\leq\frac{2c_1}{1-\mu}=c_3(0)\eta^{\alpha+1}$$since $\mu<1$.
>      - Suppose that for the $k$-th step $$||\varepsilon_k||\leq c_2(k)\eta^{\alpha},\ ||\varepsilon_{k+1}-\varepsilon_k||\leq c_3(k)\eta^{\alpha+1}.$$Then for the $(k+1)$-th step, we have $$\begin{aligned}||\varepsilon_{k+1}||&\leq||\varepsilon_{k+1}-\varepsilon_k ||+||\varepsilon_k||\\\\ & \leq c_2(k)\left(1+\eta\frac{c_3(k)}{c_2(k)}\right)\eta^{\alpha}\\\\ &\leq c_2(k)e^{\frac{2\lambda \eta}{1-\mu}}\eta^{\alpha}=c_2(k+1)\eta^{\alpha}\end{aligned}$$where the last inequality is because $e^x>1+x$ for $x>0$. Similarly, according to Eq.(a.4), $$\begin{aligned}||\varepsilon_{k+2}-\varepsilon_{k+1}||& \leq\mu||\varepsilon_{k+1}-\varepsilon_{k}||+\eta\lambda||\varepsilon_{k+1}||+c_1\eta^{\alpha+1}\\\\ &\leq\left[\mu c_3(k)+\lambda c_2(k+1)+c_1\right]\eta^{\alpha + 1}\\\\ &=\left[\mu e^{-\frac{2\lambda\eta}{1-\mu}}+\frac{1-\mu}{2}+\frac{1-\mu}{2}e^{-\frac{2\lambda\eta}{1-\mu}(k +1)}\right]c_3(k+1)\eta^{\alpha+1}\\\\&\leq\left[\mu+\frac{1-\mu}{2}+\frac{1-\mu}{2}\right]c_3(k+1)\eta^{\alpha+1}=c_3(k+1)\eta^{\alpha+1}.\end{aligned}$$Hence, for a fixed time $T = N\eta$ such that $N = \lfloor\frac{T}{\eta}\rfloor$, at the $k$-th step, the discretization error is bounded by $$||\varepsilon_k||\leq \frac{c_1}{\lambda}e^{\frac{2\lambda}{1-\mu}k\eta}\eta^{\alpha}.$$
> 2. **Comment**: *"(Eq. (17) (18))... case?"*
>
>    **Response**: As discussed earlier, as long as $\psi=\eta/(1-\mu)^2$ is small, HBF can still track the learning dynamics. However, if $\mu\approx1$ while $\eta$ is also large, though HBF may not be valid, the training dynamics of the discrete HB is not solely governed by the gradient either, as it will be largely affected by the gradient and the momentum in the current step simultaneously and it might even not converge. Hence, we believe the failure of HBF in such case is acceptable.
> 3. We agree with the reviewer that existing continuous-time formulations of Adam are quite complex, hence studying the continuous approximations for Adam should be an important future direction.
>
>     The main challenge, from our perspective, lies in that Adam incorporates momentum and adaptive learning rate at the same time. According to our approach, there will be two key steps for the problem:
>     - find the equations for the counter term $\gamma_k$ (similarly, $G_k$ should degenerate to sign gradient, the lowest-order continuous approximation of Adam);
>     - solve the equation obtained in the last step in the series form.
>
>     Hence, mathematically, given the first order ODE $$\dot{\beta}=-G_k-\eta\gamma_k,$$one difficulty lies in establishing the equations that must be satisfied by $\gamma_k$ from requiring $\varepsilon_{k+1}-\varepsilon_k=\mathcal{O}(\eta^{\alpha+1})$ (this condition will be sufficient for us to show $\varepsilon_k=\mathcal{O}(\eta^{\alpha})$) for Adam, as the formulations of Adam are more complicated compared to HB. In addition, we believe solving the equations obtained in the last step would also be challenging for Adam. Once these two challenges are resolved, the continuous approximations for Adam can also be established.
> 4. As the current approach relies on Taylor expansion to the first order of $\eta$ with remainder term in the integral form, it cannot be directly generalized to the case of arbitrary $\alpha>0.$
> 5. Yes. This is because HB utilizes the update history while GD does not.
> 6. Please refer to our response to the second point of the Weaknesses part.
> 7. We run our experiments on a Linux platform with an Intel(R) Xeon 626 CPU E5-2683 and an NVIDIA Tesla A100.
>    - For Fig.1, the running time is about 1 hour and 54 minutes.
>    - For Fig.2, the running time depends on $\eta/\eta_{\text{euler}}$. For $\eta/\eta_{\text{euler}}=10$, the experiments take about 2 minutes.
>
> ---
> Reference
>
> Leclerc and Madry. The two regimes of network training, 2020.

---

> > ### Comment · Reviewer_iEY4 · 2025-08-03
> >
> > Thank you for the thorough response, which addresses all of my concerns.
> > I kindly encourage the authors to incorporate these discussions into the revision.
> >
> > I have also read other reviews and responses and found that most concerns raised by other reviewers have been resolved.
> > Furthermore, the proof presented in the paper has been improved in the rebuttal.
> >
> > I support the acceptance of this paper.

---

> > > ### Comment · Reviewer_iEY4 · 2025-08-09
> > >
> > > As the discussion period wraps up and the discussion seems to be converging, I have added my Final Justification.
> > > Good luck.

---

> > > > ### Author Response · Authors · 2025-08-09
> > > >
> > > > We sincerely thank the reviewer for their insightful feedback and constructive comments throughout the review process, and we deeply appreciate their endorsement.

---

### Official Review · Reviewer_PYzD · 2025-06-16

**Clarity:** 4
**Significance:** 3
**Originality:** 3
**Rating:** 4
**Confidence:** 4

**Summary:**

This paper proposes a new continuous ODE model that can reduce the discretization error from the discrete HB method.
By ultilizing the Taylor expansion and discrete matching on the learning rate, the discretization error can be arbittrary order of the learning rate. This result provides a reliable foundation for analyzing the momentum methods in the continuous time limit.

**Questions:**

1.see the weakness.

2.How can the theoretical insights from Theorem 2.1 be used to improve practical optimizer design?

**Ethical Concerns:**

["NO or VERY MINOR ethics concerns only"]

**Final Justification:**

Most of my questions have been resolved. I would like to see more experiments to support the theoretical validity of the work. So I  maintain the positive score.

**Limitations:**

yes

**Quality:**

3

**Strengths And Weaknesses:**

The theoretical results are very attractive, and can be considered as a good extension of previous works.
While (Kovachki and Stuart, 2020) and (Ghosh et al., 2023) explicitly anaylzed the discretization error to $\mathcal{O}(\eta)$ and $\mathcal{O}(\eta^2)$, this paper extended the result to $\mathcal{O}(\eta^{\alpha})$ with arbitrary positive integer $\alpha>0$.
This continuous model is applied to analyze the regularization term for the directional smoothness and the implicit bias of diagonal linear networks.
This bridge the gap between the discrete HB and the its precise characterization in continuous time.
Although I did not check all the proofs in detail, I believe the overview of the proof is correct.
The following parts could be improved a bit more:

1. The numerical experiments are far from enough to reflect the correctness of the main Theorem 2.1. This paper only conduct results on 2d-simple model and 2-layer diagonal linear networks for simple regression problems. More experiments on deep learning taskes should be explored to examing the discretization errors. What I expect to see is the discretization errors vs iterations on different $\alpha$, like the Figure 2(b) in your paper (but in different tasks), or Figure3-4 in Miyagawa (2023).

2. The mathematical results Theorem 2.1 are got by ultilizing the Taylor expansion and discrete matching on the learning rate used in (Miyagawa ,2023), (Rosca et al., 2023) for GD, and extended the techniques to HB, which covers the theoretical results of previous works  (Kovachki and Stuart, 2020) and (Ghosh et al., 2023). From this point of view, the theoretical contributions are limited.

3. I have some difficulties when I first read the paper and before searching the cited works. I think the reason is that the "overview of our approch" in page 3 full fill of mathematical symbols without explicit explaning, and the following proofs are based  on the overview. The motivation and main technniques in this overview should be well-organized.

4. Some typos: lacking $\eta$ for the discrete GD in Table 1, lacking $\frac{\nabla L \nabla^2 L}{2}$ in equation (20).

---

> ### Author Rebuttal · Authors · 2025-07-31
>
> We thank the reviewer very much for the constructive comments and suggestions. We will make the corresponding improvements as suggested by the reviewer in the revision. In particular:
> 1. **Comment**: *"The numerical experiments ... in Miyagawa (2023)."*
>
>    **Response**: To address this concern, during this rebuttal period, we have conducted experiments similar to those in Fig.2(b) in the MNIST dataset, where we now train a three-layer fully-connected neural networks (FCNN) similar to that in Miyagawa (2023). The FCNN has a structure of $\texttt{Linear}(784\times128)\to \texttt{SiLU} \to \texttt{Linear}(128\times128) \to \texttt{BatchNormalization}\to \texttt{Linear}(128\times10)$. Cross-entropy is used for the loss function. The batch size is 60,000, i.e., we perform full-batch training. We let the momentum factor $\mu\in\\{0.7, 0.8\\}$ and take the learning rate as $\eta=0.01$. The results well align with that for the toy model in Fig.2(b): HBF with $\alpha = 3$ has lower discretization error compared to HBF with $\alpha = 2$, and both of them are better than the RGF. We will add these experiments in the revision, as in this stage we are not allowed to upload such results.
> 2. **Comment**: *"The mathematical ... are limited."*
>
>    **Response**: We would like to highlight that our approach, while inspired partly by  prior works, bridge an important gap of establishing arbitrary order approximations of momentum methods and can be further developed as a general framework to include more optimization algorithms. In particular, compared to results for GD, our theoretical results successfully considered one important aspect of modern optimization methods--momentum--in the continuous time model. This extension is nontrivial. For example, methods with momentum explicitly incorporate effects of historical iterations while GD does not, hence the extension to momentum methods needs to carefully consider how to characterize such effects.
>
>     In addition, approaches in Kovachki and Stuart, (2020) and Ghosh et al., (2023) require specifying the desired order of the discretization error before performing the analysis, hence are not suitable to be generalized to the case with $\mathcal{O}(\eta^{\alpha})$ for arbitrary $\alpha > 0$. As a comparison, our approach bridge this gap by considering an arbitrary $\alpha$ at the first place.
>
>    Therefore, our results provide a viable way to develop a general framework for studying high-order continuous time models for a variety of advanced optimization methods that also utilize momentum such as Adam. Hence our contributions could be significant.
>
> 3. **Comment**: *"I have some difficulties ... well-organized."*
>
>    **Response**: We thank the reviewer for the suggestion. In the revision before presenting the discussion of "overview of our approach" in page 3, we will add a new paragraph to provide a more clear and concise verbal explanation of the motivation for our approach. This organization allows readers to first understand the motivation behind our approach before exploring the technical aspects.
>
>    **Paragraph that will be added to the revision** "Given the discrete HB method, our overall goal is to find the formulation of a modified ODE, $$
>     \dot{\beta} = - G_k - \eta \gamma_k, \tag{a.1} $$
>     such that the discretization error can be controlled arbitrarily low, where the design of the ODE Eq.(a.1) follows two basic principles:
>     - When $\eta$ is very small, Eq.(a.1) should coincide with the simplest continuous approximation of HB, the rescaled gradient flow. Hence, $G_k$ should be able to degenerate to the rescaled gradient $\nabla L / (1 - \mu)$.
>     - By adding the counter term $\gamma_k$, Eq.(a.1) should allow us to further decrease the discretization error of the rescaled gradient flow to $\mathcal{O}(\eta^{\alpha})$ for any $\alpha > 0$ to get better continuous approximations for HB.
>
>     With Eq.(a.1), it is left for us to derive the exact forms of $G_k$ and $\gamma_k$. For this purpose, there will be two key steps:
>     - Find the equations that $G_k$ and $\gamma_k$ must satisfy if we require the discretization error $\varepsilon_k = \mathcal{O}(\eta^{\alpha})$ for any $\alpha > 0$: this is achieved by (i). deriving the formulation of $\varepsilon_{k + 1} - \varepsilon_k$; (ii). requiring $\varepsilon_{k + 1} - \varepsilon_k = \mathcal{O}(\eta^{\alpha + 1})$, as this condition will be sufficient for us to prove $\varepsilon_k = \mathcal{O}(\eta^{\alpha})$ hence can give us the corresponding equations for $G_k$ and $\gamma_k$, respectively.
>     - Solve these equations to give the formulations of $G_k$ and $\gamma_k$: deriving $G_k$ will be straightforward, and the construction of $\gamma_k$ will be done by carefully expressing  the corresponding equation obtained in the first step as an explicit functional for $\gamma_k$ using the series form, which can then be solved by matching terms in each order of $\eta$."
> 4. Thanks for pointing the typos out. We will correct this in the revision.
>
> ---
>
> ### Questions Part
>
> **Comment**: *"How can the ... optimizer design?"*
>
>    **Response**: One interesting observation according to HBF with $\alpha = 3$ is its implicit preference of low directional smoothness, and this preference can be explicitly added to optimizers as a regularization to improve the performance, since low directional smoothness typically is related to low sharpness and better generalization performance.
>
> ---
>
> Reference
>
> Kovachki and Stuart. Continuous time analysis of momentum methods.
>
> Ghosh et al. Implicit regularization in heavy-ball momentum accelerated stochastic gradient descent.
>
> Miyagawa. Toward equation of motion for deep neural networks: Continuous-time gradient descent and discretization error analysis.

---

> > ### Comment · Reviewer_PYzD · 2025-08-02
> >
> > Thanks for the detailed responses. Most of my questions have been resolved. I would like to see more experiments to support the theoretical validity of the work. For now, I will maintain my positive score.

---

### Official Review · Reviewer_YrV8 · 2025-07-01

**Clarity:** 2
**Significance:** 2
**Originality:** 2
**Rating:** 4
**Confidence:** 3

**Summary:**

This work proposes a continuous time framework--HB Flow (HBF)--for the heavy-ball momentum method (HB). The authors show HBF approximate HB up to arbitrary order. They also study the implicit bias of HBF on diagonal linear networks which brings insights into the implicit bias problem in momentum methods. Experiments on simple neural networks are done to verify their results.

**Questions:**

1. In Figure 2(a), I noticed that the $\alpha=2$ case gives a better approximation away from the convergence point while $\alpha=3$ is the opposite. I wonder if the authors have some intuition for this behavior.

2. This analysis can be generalized to the stochastic case by for example replacing $\nabla L$
 by $\tilde{\nabla} L$
 where $\tilde{\nabla}$
 denotes the approximate gradient. The approximation can be done by either introducing a diffusion term [1] or an index switching process [2]. I think this might be an interesting future direction.

[1] Stochastic modified equations and dynamics of stochastic gradient algorithms I: Mathematical foundations, Li et al.

[2] Analysis of Stochastic Gradient Descent in Continuous Time, Latz.

**Ethical Concerns:**

["NO or VERY MINOR ethics concerns only"]

**Final Justification:**

all concerns are addressed.

**Limitations:**

yes

**Quality:**

3

**Strengths And Weaknesses:**

*I have reviewed this paper last year. I don't see many significant changes. So I am borrowing some of my previous review here, which the authors had not yet addressed.*

Strengths.

1. The authors propose HB flow--a continuous-time approximation of the heavy ball momentum method. By adding back the discretization error in each time interval, the HB flow achieves an arbitrarily close distance to the discrete-time dynamic. This is justified rigorously in theorem 2.1

2. More precisely, the distance between two dynamics is measured using $L^\infty[0, T]$ on the trajectory space. The $L^\infty$ is qualified by an upper bound in the learning rate $\eta^\alpha$. The authors give the specific form for the case $\alpha = 1, 2, 3$ as examples, which help the readers to understand the construction of the continuous dynamic.

3. The authors computed the implicit bias of HBF for diagonal linear networks in theorem 3.1 which makes this framework useful in studying the generalization properties of HB. This type of behavior cannot be captured by lower order models like RGF.

4. In numerical experiments, Figure 2 illustrate the trajectories in a clear way and verifies main results from the paper.

5. I see the authors added one experiments about the implicit regularization.

Weaknesses.

1. I found the main theorem hard to follow. Specifically, Theorem 2.1 is stated with many notations without enough intuition/explanations. I am trying to understand this at a high level. So basically find the difference between continuous and discrete dynamics and use Taylor expansion for it. The order $\alpha$ in $\eta^\alpha$ just means how many times we do Taylor expansion in $t$ because the time interval is of length $\eta$. Although the computation needed to find the exact expressions is not trivial, one could argue this is not mathematically sophisticated.

2. Some notations are not consistently defined. For example, on page 4, the authors stated $G_k \in R^d$, i.e. a d dimensional vector, but $G_k$ is a function. Similar situation for $\gamma_k$.

---

> ### Author Rebuttal · Authors · 2025-07-31
>
> We thank the reviewer very much for the valuable comments. Below we address your concerns.
>
> ---
>
> ## Our Improvements
> We would like to first highlight that we have made **a series of important improvements** in this version:
>
> 1. We have added some paragraphs to help the readers to understand the merit of our approach, e.g., the overview of our approach in line 128-147, page 3.
> 2. We have improved the statement of Theorem 2.1, where now we:
>    - first state what the formulation of the proposed ODE with discretization error to arbitrary order of $\eta$ will be;
>    - then state what the explicit formulations of the components of the ODE will be by defining necessary notations.
> 3. As noted by the reviewer, we have also added new experiments about the implicit regularization for the directional smoothness of HB, which is a novel result and reveals the difference compared GD. This clearly shows the significance of studying continuous time model with lower discretization order.
>
> ---
>
> ## Weaknesses Part
> Below we address your concerns mentioned in the Weaknesses part.
>
> 1. **Comment**: *"I found the main theorem ... mathematically sophisticated."*
>
>    **Response**: We would like to first clarify that our results  are not obtained by performing Taylor expansion to the specific order of $\eta$. Instead, our approach performs Taylor expansion with remainder term to the first order. Meanwhile, our approach involves a delicate construction of the modified ODE such that we are always able to build continuous time models with discretization errors to arbitrarily high order of $\eta$ by directly employing existing lower order ones, rather than performing the whole set of analysis. This cannot be achieved by performing Taylor expansion to a specific order in advance.
>
>     In addition, our result builds a continuous time model for HB with discretization error to arbitrary order of $\eta$ for the first time, making it a significant one. Moreover, our approach might also be generalized to establish higher-order continuous time models for more optimization methods, e.g., Adam. This is because the intuition behind our approach is natural  and effective, and we do not think this is a downside of our approach to make it less mathematically sophisticated.
>
>     We now address your concerns. On one hand, the notations in Theorem 2.1 are unavoidable due to the fact that higher-order continuous approximations typically require a more delicate construction. On the other hand, we will further add more explanations and intuitive illustration for our approach at a high level (please find the explanation of our intuition below) as suggested by the reviewer in the revision.
>
>     **Intuition:** Overall, for the discrete HB method, our goal is to find a counter term $\gamma_k$ for the modified ODE proposed by us $$
>     \dot{\beta} = - G_k - \eta \gamma_k, $$
>     such that the discretization error can be controlled as $\varepsilon_k = \mathcal{O}(\eta^{\alpha})$. To achieve this, mathematically, we need to derive two sets of equations for $\gamma_k$ and $G_k$, respectively, that must be satisfied under the condition $\varepsilon_k = \mathcal{O}(\eta^{\alpha})$. Then these equations can be solved to give us the desired formulations for $\gamma_k$ and $G_k$, respectively. Therefore, there are two crucial parts of our approach:
>
>     - Find the equations that $G_k$ and $\gamma_k$ must satisfy if we require $\varepsilon_k = \mathcal{O}(\eta^{\alpha})$: this is achieved by (i). deriving the formulation of $\varepsilon_{k + 1} - \varepsilon_k$ by Taylor expanding of $\beta(t_{k\pm 1})$ at $t = t_k$ to the first order of $\eta$ with the remainder term in the integral form; (ii). requiring $\varepsilon_{k + 1} - \varepsilon_k = \mathcal{O}(\eta^{\alpha + 1})$, which will be sufficient for us to prove $\varepsilon_k = \mathcal{O}(\eta)$ hence can give us the corresponding equations for $G_k$ and $\gamma_k$, respectively.
>     - Solve these equations to give the formulations of $G_k$ and $\gamma_k$: deriving $G_k$ is straightforward, and the construction of $\gamma_k$ is done by carefully deriving $I_k$ as an explicit functional for $\gamma_k$ such that both of them will be in the series form, then the corresponding equation can be solved by matching terms in each order of $\eta$.
> 2. **Comment**: *"Some notations ... Similar situation for $\gamma_k$"*
>
>    **Response**: We thank the reviewer for pointing this. For general $\beta\in \mathbb{R}^{d}$, $G_k$ and $\gamma_k$ are functions (vector fields). We will correct this in Page 4. At a specific point of $\beta$, $G_k$ and $\gamma_k$ will be $d$-dimensional vectors.
>
> ---
>
> ## Questions Part
> 1. **Comment**: *"In Figure ..."*
>
>    **Response**: Intuitively, continuous approximations with discretization error $\varepsilon_k = \mathcal{O}(\eta^{\alpha})$ with $\alpha > 1$ are obtained by adding counter terms to that with $\alpha = 1$, hence these counter terms will lead their trajectories to intersect the trajectory of discrete HB during the early stage. In particular, the intersection for HBF with $\alpha = 2$ occurs in the early stage, making its performance appear to be better. In addition, although the trajectory is long in the early stage, the points are sparsely distributed as the gradient is large at this stage. Therefore, when evaluating the overall discretization error across all points, HBF with $\alpha = 3$ tracks the discrete HB better, e.g., it demonstrates the lowest discretization error very quickly (after only 10 steps according to Fig.2(b)).
> 2. **Comment**: *"This analysis can be generalized ... future direction."*
>
>    **Response**: We thank the reviewer for the insightful suggestion. We agree with the reviewer that incorporating stochasticity by replacing $\nabla L$ with $\tilde{\nabla} L$ is an interesting direction, and we believe it might be more interesting if our approach can be generalized to include the stochasticity for the more advanced optimizer Adam. We will add a discussion of the extension to stochastic HB by applying techniques from these suggested works in the revision.

---

> > ### Comment · Reviewer_YrV8 · 2025-08-03
> >
> > The authors have addressed all my concerns. I have changed my rating to 4. Good luck.

---

> > > ### Author Response · Authors · 2025-08-09
> > >
> > > We sincerely thank the reviewer for their insightful feedback and constructive comments throughout the review process, and we deeply appreciate their endorsement.

---

### Official Review · Reviewer_nQN7 · 2025-07-02

**Clarity:** 2
**Significance:** 2
**Originality:** 3
**Rating:** 4
**Confidence:** 4

**Summary:**

In this paper, the authors propose a continuous ode approximation of the popular Heavy-ball momentum optimization algorithm. Compared to the previous studies, the framework proposed in this paper can achieve an error as small as arbitrary order of the small interpolation, i.e. the learning rate $\eta$. Based on this conclusion, authors propose a new implicit bias of diagonal linear networks. While simple and idealized, this model usually serves as the first step for multiple theoretical studies.

**Questions:**

While not a major issue, as a study including the implicit bias of momentum, I suggest the author could consider discussing [1, 2, 3] in the related works.

[1]. Wang et al. The implicit bias for adaptive optimization algorithms on homogeneous neural networks. ICML.

[2]. Wang et al. Does Momentum Change the Implicit Regularization on Separable Data? Neurips.

[3]. Zhang et al. The implicit bias of Adam on separable data. Neurips.

**Ethical Concerns:**

["NO or VERY MINOR ethics concerns only"]

**Final Justification:**

The rebuttal addresses my concerns. I have updated my score to 4.

**Limitations:**

The potential incorrectness in the proof by using the notation of $O(\cdot)$ in induction.

**Paper Formatting Concerns:**

No.

**Quality:**

2

**Strengths And Weaknesses:**

**Strength:**

1. Compared to the previous studies, the framework proposed in this work can better capture the optimization trajectories of the Heavy-ball momentum, as it can achieve an arbitrarily small error, which, also claimed as the major technical contribution of this work.

2. The study of the implicit bias of momentum implies the potential benefits of studying the the higher-oder approximations.

**Weaknesses:**

1. While I found the approximation w.r.t. an arbitrary order of the interpolation size $\eta$ is relatively interesting, the intuition behind such an approach is hard to follow, especially after reading the proof details in the Appendix. First of all, from my understanding, the presentation of equation (6) is quite different from the standard form of second-order Taylor expansions. Indeed, after checking the definition of $\eta^2 I_k^+$ in line 140, it is exactly $\beta(t_{k+1}) - \beta(t_{k}) - \eta\dot{\beta}(t_{k}^+)$. Therefore, from my perspective, term $I_k^+$ is entirely determined by $\beta(\cdot)$ and is fixed through the entire stage. However, from the proof and the discussion of the proof, it seems that $I_k^+$ is **specifically designed** to match the term of $\gamma$. Besides, what is the advantage to utilize the calculation of equation (47)? It seems that it still introduces the $n$-th order differential calculations here. I wonder more intuitive explanation of such a calculation. In addition, I hope that authors can also clarify some of the notations. For example, is the domain of $G_k(\beta)$ only from $\beta_k$ to $\beta_{k+1}$, or the entire space? If it is the former, then $G_k(\beta)$ would always be a multiple of $\nabla L(\beta)$, in which case it is unclear what is gained by defining distinct $G_k(\beta)$ functions. If it is the latter, then how should we interpret the expression $G_{k-1}(\beta_{t_k})$ in equation (44)? Moreover, when using the shorthand $G_k$, what is the intended independent variable?

2. The use of the notation $O(\cdot)$ can lead to significant incorrectness in induction and should therefore be avoided in such contexts. To illustrate what I claim, let's consider a classical paradox when using the $O(\cdot)$ in induction. For any integer $K$, it is obvious that we can write $K=\sum_{k=1}^K a_k$, with $a_k=1$. Clearly, $a_1 = O(1)$ holds. Next, assume that at the $k'$-th iteration $\sum_{k=1}^{k'} a_k = O(1)$, which leads to the conclusion that $\sum_{k=1}^{k'+1} a_k = O(1) + 1 = O(1)$. Consequently, for any integer, no matter how large it is, we can always claim that it is $O(1)$, which, is obviously incorrect. To make induction mathematically rigorous, authors should explicitly specify the universal absolute constant hided in $O(\cdot)$, and keep such a constant fixed during the proof.


I would like to increase my score if the authors could provide further illustration and modify their proof (especially the $O(\cdot)$ notation used in their induction proof of Lemma A.2.)

---

> ### Author Rebuttal · Authors · 2025-07-31
>
> We thank the reviewer very much for the insightful comments. We appreciate this opportunity to address your concerns.
>
> ---
> ## Weaknesses Part
>
> ### Weaknesses 1
> We address concerns in Weaknesses 1 point by point.
> - **Comment**: *"While I...Appendix."*
>
>    **Response**: To better illustrate the intuition of our approach, we first summarize it below. Overall, our goal is to find the formulation of the counter term $\gamma_k$ for the ODE $$\dot{\beta}=-G_k-\eta\gamma_k\tag{a.1}$$ such that the discretization error can be controlled as $\varepsilon_k = O(\eta^{\alpha})$. To achieve this, our approach aims to find a functional equation for $\gamma_k$ that must be satisfied under the condition $\varepsilon_k = O(\eta^{\alpha})$, and this functional equation can be solved to give the desired $\gamma_k$.
> - **Comment**: *“First of...term of $\gamma$”*
>
>    **Response**: Eq.(6) is the first-order Taylor expansion with the remainder term $I_k$ in the integral form. $I_k$ is determined by $\gamma_k$ as noted by the reviewer, and it is not specifically designed to match $\gamma_k$. Instead, it is $\gamma_k$ that is specifically designed to solve the functional equation mentioned above by matching terms.
>
>    To illustrate this in detail, below we indicate how we establish and solve such a functional equation. Given the unknown $\gamma_k$ in Eq.(a.1), our method follows a general flow:
>     $$\begin{aligned}\boxed{\text{Step 1}}\quad\text{unknown }&\gamma_k\overset{\text{determines}}{\longrightarrow}I_k^{\pm}\\\\&\ \searrow\qquad\ \swarrow\\\\\boxed{\text{Step 2}}\quad\ \ \ \text{Taylor }&\text{Expansion of }\beta(t_{k\pm1})\\\\&\qquad\ \downarrow\\\\ \boxed{\text{Step 3}}\quad\quad\text{Expres}&\text{sion of }\varepsilon_{k+1}-\varepsilon_k\\\\&\qquad\ \downarrow\text{\small required to be }O(\eta^{\alpha+1})\text{ \small to guarantee }\varepsilon_k=O(\eta^{\alpha})\\\\
>    \boxed{\text{Step 4}}\quad\ \text{Function}&\text{al equation for }\gamma_k\\\\&\qquad\ \downarrow\text{\small solved by matching to each order of }\eta\\\\
>     \boxed{\text{Step 5}}\qquad\qquad\quad&\text{Solution of }\gamma_k
>     \end{aligned}$$
>    Specifically,
>     - In Step 1, as noted by the reviewer, given arbitrary unknown $\gamma_k$ in Eq.(a.1), $I_k^{\pm}$ is then determined, which is also unknown but depends on $\gamma_k$ (e.g., their relation is constructed in Lemma A.1 (line 463-466, page 13) if $\gamma_k$ is a series)
>     - In Step 2, given the unknown $\gamma_k$ and the corresponding $I_k^{\pm}$, we can derive the Taylor expansion for $\beta(t_{k\pm1})$ (see Eq.(6) in line 138-139), which depends on both $\gamma_k$ and $I_k^{\pm}$.
>     - In Step 3, given the Taylor expansions in Step 2, we can construct a set of expressions for the discretization error $\varepsilon_{k+1}-\varepsilon_k$ (Eq.(7) in line 141-142 page 4).
>     - In Step 4, to guarantee $\varepsilon_k = O\eta^{\alpha})$, the condition $\varepsilon_{k + 1} - \varepsilon_k = O(\eta^{\alpha + 1})$ is sufficient (this statement will be formally proved in Lemma A.2 (line 477-481 page 14)), which can be satisfied by letting $$
>     I_k^++\mu I_k^--\gamma_k+\mu\gamma_{k-1}=O(\eta^{\alpha-1})\tag{a.2} $$
>     in Eq.(34) (line 454-455, page 13).
>     - In Step 5, as $I_k^{\pm}$ depends on $\gamma_k$, Eq.(a.2) is in fact a functional equation for $\gamma_k$. To solve this equation, we write $\gamma_k$ in the series form $$\gamma_k=\sum_{\sigma}^{\infty}\eta^{\sigma}\gamma_k^{(\sigma)}, $$
>     then Lemma A.1 provides us the corresponding $I_k^{\pm}$, which is also in the series form:
>     $$I_k^{\pm}=\sum_{\sigma}^{\infty}\eta^{\sigma}(\mathscr{I}_k^{\pm})^{(\sigma)},$$ where each $(\mathscr{I}_k^{\pm})^{(\sigma)}$ is determined by the lower-order terms of $\gamma_k$. If we match $\gamma_k-\mu\gamma _ {k-1}$ with $I_k^{+}+\mu I_k^{-}$ to each order of $\eta$, then we obtain a set of conditions that must be satisfied by $\gamma_k^{(\sigma)}$ for any $\sigma$, which can be solved to give the forms of $\gamma_k^{(\sigma)}$.Then a simple truncation of $\gamma_k$ to the $(\alpha-2)$-th order, i.e.,
>
>       $$\gamma_k = \sum_{\sigma}^{\alpha - 2} \eta^{\sigma} \gamma_k^{(\sigma)},$$ guarantees Eq.(a.2).
> - **Comment**: *“Besides,...calculation.”*
>
>    **Response**: We first confirm that Eq.(47) still introduces $n$-th order differential calculation. The advantage of Eq.(47) is as follows.
>
>    Intuitively, we aim to derive the formulation of $I_k$ in the series form such that the functional equation Eq.(a.2) can be solved by matching terms to different orders of $\eta$. Hence, we need to group together terms of the same order in $\eta$ in $d^n\beta / dt^n$ (as $I_k$ is determined by $d^n\beta / dt^n$). The advantage of Eq.(47) is that the operator, $(\sum_{\sigma = 0}^{\infty} \eta^{\sigma}\gamma_k^{(\sigma)})\cdot\nabla$, depends on $\eta$ explicitly, allowing us to gather terms in the same order easily.
> - **Comment**: *"In addition,...equation (44)?"*
>
>    **Response**: We first confirm that the domain of $G_k$ is the entire space. Below we answer the questions separately.
>    - The reason why we use the notation $G_k$ is to indicate that we expect it to be dependent on the iteration count $k$ explicitly in the modified equation Eq.(a.1).
>     - Interpret Eq.(44): We would like to first mention that $G_{k-1}(\beta_{t_k})$ in Eq.(44) is a typo, and it should be $G_{k-1}(\beta(t_k))$. We apologize for potential misunderstanding due to this typo. As mentioned earlier, the condition $\varepsilon_{k + 1} - \varepsilon_k=O(\eta^{\alpha+1})$ is sufficient to guarantee $\varepsilon_k = O(\eta^{\alpha})$. To make this condition satisfied at $\beta(t_k)$, besides the functional equation Eq.(a.2), we require a general recursive relation for $G_k$ in the entire space of $\beta$: $$\forall k\in [N]:G_k(\beta)=\mu G_{k-1}(\beta)+\nabla L(\beta),\tag{a.3}$$which implies $$\forall k\in [N]:G_k(\beta)=\frac{1-\mu^{k+1}}{1-\mu}\nabla L(\beta)\tag{a.4} $$in Eq.(16) (line179-180, page 5). As the domain of $G_k(\beta)$ is the entire space, Eq.(a.3) is also satisfied at $\beta(t_k)$, which is exactly Eq.(44). Thus, along with the functional equation Eq.(a.2), $\varepsilon_{k + 1} - \varepsilon_k = O(\eta^{\alpha + 1})$ can be guaranteed. For large $k$, $G_k$ becomes the rescaled gradient $\nabla L/(1-\mu)$.
>
>    - The use of $G_k$: When using the shorthand $G_k$, the intended independent variable is $\beta$, i.e., it is a shorthand of $G_k(\beta)$ and closely related to gradient via Eq.(a.4). The subscript $k$ only indicates that $G_k$ will change for different iteration count $k$, and this dependence on $k$ will be negligible for large $k$.
> ### Weaknesses 2
> - **Comment**: *“The use...proof.”*
>
>    **Response**: We thank the reviewer for this insightful suggestion. We provide our proof below, where we specify the universal absolute constant in the $O$ notation explicitly as suggested by the reviewer.
>
>     We first present several useful relations. As Eq.(43) is established by solving the functional equation for any iteration count $k$, we can write the relation between $\varepsilon_{k + 1}$ and $\varepsilon_{k}$ (Eq.(34), line454-455 in page 13) as $$\varepsilon_{k+1}-\varepsilon_{k}=\mu(\varepsilon_{k}-\varepsilon_{k-1})-\eta[\nabla L(\beta(t_k))-\nabla L(\beta(t_k)-\varepsilon_{k})]+O(\eta^{\alpha + 1}).\tag{a.5}$$If $\varepsilon_{k}=O(\eta^{\alpha})$, then Eq.(a.5) implies $$||\varepsilon_{k+1}-\varepsilon_{k}||\leq\mu||\varepsilon_{k}-\varepsilon_{k-1}||+\eta\lambda||\varepsilon_{k}||+c_1\eta^{\alpha+1}$$for some constant $c_1$ where we use $G_k-\mu G_{k-1}=\nabla L$ and let $\lambda=\max_{\beta}||\nabla^2L(\beta)||$. Denoting$$\forall k:c_2(k)=\frac{c_1}{\lambda}e^{\frac{2\lambda \eta}{1-\mu}k},\ c_3(k)=\frac{2c_1}{1-\mu}e^{\frac{2\lambda\eta}{1-\mu}k},$$we now prove by induction.
>     - For the first step ($k = 0$), by definition we have $$\varepsilon_0=0\leq c_2(0)\eta^{\alpha}.$$
>     Note that $G_{-1}=0$ and $\gamma_{-1}=0$ by definition, we have $$||\varepsilon_1-\varepsilon_0||\leq c_1\eta^{\alpha+1}\leq\frac{2c_1}{1-\mu}=c_3(0)\eta^{\alpha+1}$$since $\mu<1$.
>     - Suppose that for the $k$-th step $$||\varepsilon_k||\leq c_2(k)\eta^{\alpha},\ ||\varepsilon_{k+1}-\varepsilon_k||\leq c_3(k)\eta^{\alpha+1}.$$ Then for the $(k + 1)$-th step, we have $$\begin{aligned}||\varepsilon_{k+1}||&\leq||\varepsilon_{k+1}-\varepsilon_k||+||\varepsilon_k||\\\\&\leq c_2(k)\left(1+\eta\frac{c_3(k)}{c_2(k)}\right)\eta^{\alpha}\\\\&\leq c_2(k)e^{\frac{2\lambda \eta}{1-\mu}}\eta^{\alpha}=c_2(k+1)\eta^{\alpha}\end{aligned}$$where the last inequality is because $e^x>1+x $ for $x>0$. Similarly, according to Eq.(a.4), $$\begin{aligned}||\varepsilon_{k+2}-\varepsilon_{k+1}||&\leq\mu||\varepsilon_{k+1}-\varepsilon_{k}||+\eta\lambda||\varepsilon_{k+1}||+c_1\eta^{\alpha+1}\\\\&\leq\left[\mu c_3(k)+\lambda c_2(k+1)+c_1\right]\eta^{\alpha+1}\\\\&=\left[\mu e^{-\frac{2\lambda\eta}{1-\mu}}+\frac{1-\mu}{2}+\frac{1-\mu}{2}e^{-\frac{2\lambda\eta}{1-\mu}(k+1)}\right]c_3(k+1)\eta^{\alpha+1}\\\\&\leq\left[\mu+\frac{1-\mu}{2}+\frac{1-\mu}{2}\right]c_3(k+1)\eta^{\alpha+1}=c_3(k+1)\eta^{\alpha+1}.\end{aligned}$$This now proves our claim: for a fixed time $T = N\eta$ such that $N = \lfloor\frac{T}{\eta}\rfloor$,  we have $$\forall k\in [N]: ||\varepsilon_k||\leq\frac{c_1}{\lambda}e^{\frac{2\lambda}{1-\mu}k\eta}\eta^{\alpha}.$$
>
>      Note that we need the total time $T = N\eta = O(1)$ to control $\varepsilon_k$ globally. This is a typical requirement for studying continuous approximations with bounded global discretization error. We will clearly state this condition in the revision.
>
> Finally, we thank the reviewer for these insightful questions and suggestions, and we will update the proof and provide a more clear sketch of the corresponding intuition in the revision to improve the readability.
>
> ----
>
> ## Questions Part
> We thank the reviewer for suggesting these nice related works. We will discuss these related works in the revision.

---

> > ### Comment · Reviewer_nQN7 · 2025-08-02
> >
> > I thank the authors for their detailed reply, which helps me understand some technical details. I updated my score to 4.

---

> > > ### Author Response · Authors · 2025-08-09
> > >
> > > We sincerely thank the reviewer for their insightful feedback and constructive comments throughout the review process, and we deeply appreciate their endorsement.

---

### Decision · Program_Chairs · 2025-09-17

**Decision:**

Accept (poster)

**Comment:**

This paper was accepted because it provides a significant technical contribution by developing a more accurate continuous-time framework for the Heavy-Ball momentum method, which allows for a precise analysis of discretization error. All reviewers agreed that this was a major strength and that the resulting insights into implicit bias were valuable. While there were initial concerns about the clarity of the presentation and the sufficiency of the experiments, the authors provided a thorough rebuttal that satisfied most of these concerns, including more formal proof sketches, and new experiments. The positive response from the reviewers, including two who raised their scores, makes this a clear accept.